# What One View Reveals, Another Conceals: 3D-Consistent Visual Reasoning with LLMs

**Laszlo Freund**                                                           *freundl0509@gmail.com*
*SatiNav Robotics*

**Dan Kushnir**                                                *dan.kushnir@nokia-bell-labs.com*
*Nokia Bell Laboratories, New Jersey, USA*

**Reviewed on OpenReview:** *https://openreview.net/forum?id=XXXX*

## Abstract

Maintaining semantic label consistency across multiple views is a persistent challenge in 3D semantic object detection. Existing zero-shot approaches that combine 2D detections with vision-language features often suffer from bias toward non-descriptive viewpoints and require a fixed label list to operate on. We propose a truly open-vocabulary algorithm that uses large language model (LLM) reasoning to relabel multi-view detections, mitigating errors from poor, ambiguous viewpoints and occlusions. Our method actively samples informative views based on feature diversity and uncertainty, generates new label hypotheses via LLM reasoning, and recomputes confidences to build a spatial-semantic representation of objects. Experiments on controlled single-object and multi-object scenes show double digit improvement in accuracy and sampling rate over ubiquitous fusion methods using YOLO, CLIP, and other LLM-based baselines. We demonstrate in multiple settings that **L**LM-guided **A**ctive **D**etection and **R**easoning (LADR) balances detail preservation with reduced ambiguity and low sampling rate. We provide theoretical convergence analysis showing exponential convergence to a correct and stable semantic label.

## 1 Introduction

Consistently detecting objects across multiple viewpoints is a crucial task for autonomous agents, such as drones and robots. A single object may appear vastly different depending on the viewpoint, lighting, or degree of occlusion, and visual features extracted from such views often drift in embedding space. As a result, inconsistent labels emerge when fusing detections across views, leading to degraded spatial-semantic representations and downstream performance.

Recent zero-shot approaches (Jatavallabhula et al., 2023; Peng et al., 2023; Cartillier et al., 2021), address this by combining off-the-shelf detectors (Redmon et al., 2016) with vision-language models (Radford et al., 2021; Cherti et al., 2023) to assign open-vocabulary labels in 3D. While these methods avoid task-specific retraining, they rely heavily on two components: (1) the accuracy of the underlying detector, and (2) the similarity between extracted image features and a user-defined list of candidate labels. Both dependencies introduce bottlenecks. First, misdetections or low-quality views (such as those from the back of an object) can dominate the fused feature representation, biasing the final label. Second, reliance on a user-defined list of labels limits true open-vocabulary capability, hampers generalization to novel categories, and constrains the level of detail that can be captured for each object.

We propose a different approach referred to as **LADR** (**L**LM-guided **A**ctive **D**etection and **R**easoning). LADR uses large language model (LLM) reasoning to actively refine and reweight multi-view detections. Instead of passively aggregating features, our method iteratively samples informative viewpoints based on feature diversity, prompts an LLM to generate and refine label hypotheses from available visual evidence, and recomputes label confidences accordingly. This reasoning process reduces the influence of misleading

views, removes the need for a fixed label set, and enables a more robust spatial-semantic representation of the scene. We provide rigorous Markov process-based analysis for an exponential convergence rate to consistent labels. It shows differentiation in rates on the ablated versions of LADR, proving that active uncertainty based sampling with geometric grounding is the best approach among LADR algorithms.

Our contributions are as follows:

- **LLM-guided relabeling for 3D consistency:** An open-vocabulary method that uses LLM reasoning to correct viewpoint-induced misclassifications without retraining.

- **Smart sampling strategy:** An active selection of views based on feature diversity, uncertainty estimation, and geometric grounding, balancing detail preservation with reduced context ambiguity.

- **Spatial-semantic mapping:** A representation that integrates refined labels with object geometry, suitable for downstream 3D tasks.

- **Comprehensive evaluation:** single and multi-object experiments across diverse environments showing improvement of over 40%, respectively. Experiments measure 3D semantic label accuracy and sampling rate, over ubiquitous fusion methods including YOLO, and CLIP. We also provide a real-world data set study demonstrating the advantages in LADR when coping with real-life challenges, such as limited view positions, occlusions, challenging background, and more. We also provide a wall clock running-time analysis for different VLM sizes and deployments.

- **Theoretical analysis** for each ablated LADR version, proving exponential convergence to consistent semantic labels, with increasingly stronger constants for added components.

Our contributions establish a framework for zero-shot open-vocabulary 3D understanding that combines semantic reasoning, efficient view selection, and spatial integration, leading to more robust and consistent labeling across diverse scenarios.

## 2 Related work

**Foundation Models in Object Detection.** Object detection has rapidly advanced from region-based CNNs and single-stage detectors to foundation models, which enable more general and flexible representations beyond closed-set training. Architectures such as YOLO-World and YOLOE (Cheng et al., 2024; Wang et al., 2025) leverage large-scale pretraining to improve detection accuracy and adaptability across diverse scenarios. Vision-language models (VLMs) like CLIP (Radford et al., 2021; Cherti et al., 2023) provide open-vocabulary capabilities by connecting visual features with text embeddings, while models such as Segment Anything (Kirillov et al., 2023) offer class-agnostic segmentation that can be integrated into detection pipelines. Multimodal large language models like GPT-4V (OpenAI, 2024) (and later versions) further complement these approaches by enabling zero-shot reasoning over visual inputs, making them useful for refining labels and guiding exploration. These approaches demonstrate the potential to reduce reliance on task-specific training and expand detection to previously unseen categories.

**Open-Vocabulary 3D Object Detection.** ConceptFusion (Jatavallabhula et al., 2023) builds open-vocabulary 3D object maps by combining pretrained VLMs with 3D scene representations. The method uses YOLO-World as an initial object detector and Segment Anything for segmentation, attaching VLM features (e.g., from CLIP) to 3D points reconstructed from RGB-D scans, with features from multiple 2D observations aggregated via simple averaging (which ignores the 3D consistency problem). While it aims to assign open-vocabulary labels, the object categories are ultimately constrained to a fixed set. Peng et al. (2023) takes a voxel-based approach, backprojecting per-pixel CLIP features into a 3D voxel grid and fusing multiple views using different pooling strategies (random, median, or mean) among these approaches, mean pooling yields the most stable results. Kassab et al. (2024) revisits design choices for open-vocabulary 3D labeling by selecting a single "best" view per object based on a confidence metric, with the entropy of CLIP similarities with category embeddings performing best. In contrast, LADR leverages LLM reasoning to

iteratively identify and reweight informative views, producing a more robust spatial-semantic representation that is less sensitive to viewpoint bias and not limited by a fixed label set.

Both Fu et al. (2024) and Yuheng et al. (2023) pursue language-enhanced 3D understanding, yet they address complementary aspects to LADR. Scene-LLM focuses on embedding dense 3D representations inside an LLM to enable interactive scene reasoning, and embodied tasks. OVODA, is detection-centric, extending 3D object recognition to open-vocabulary classes with rich attributes through multimodal alignment. LADR, instead, targets cross-view semantic consistency: rather than building an LLM-conditioned 3D agent or enriching detector outputs with attributes. LADR uses LLMs to actively select informative views and reconcile conflicting predictions across viewpoints to produce a 3D-consistent semantic scene interpretation.

**Active exploration.** Active exploration in embodied agents aims to optimize camera or agent trajectories to reduce uncertainty and collect informative observations. SEAL Chaplot et al. (2021) and subsequent works Scarpellini et al. (2024) introduce a self-supervised framework in which agents explore their environment to learn semantic segmentation without manual labels, leveraging 3D spatial consistency. These methods train an exploration policy to target novel or uncertain areas, optimizing coverage of diverse object views. Features from multiple viewpoints are re-projected into a shared 3D space using depth and camera poses, and a 3D consistency loss ensures that features corresponding to the same physical point remain consistent across views. This supervision enables learning of a semantic segmentation function directly from RGB-D frames, without human annotations, and replaces random or fixed path planning with informed, targeted exploration. While effective, these approaches require reinforcement learning policies and multiple rollouts, which are computationally expensive. In contrast, zero-shot LLM-based methods can reason about object semantics and its correlation with the viewpoint directly from observations without task-specific policy training, avoiding overhead and sample inefficiency inherent to learned exploration policies.

## 3 The 3D Consistency Problem

We consider the problem of receiving as input a set of images covering objects of interest, where the required output is a model that can correctly assign to each of the objects its correct label from any viewpoint it is being observed.

Achieving consistent object labeling across multiple viewpoints remains a key obstacle in 3D perception. In multi-view pipelines, each observation of an object is processed independently before being fused into a unified label. When these observations are heterogeneous (due to varying viewpoints, occlusions, or lighting) the resulting feature embeddings can drift toward non-representative appearances. This drift can overweight misleading views, leading to label instability.

In zero-shot approaches such as those combining YOLO detections with CLIP embeddings, the problem is exacerbated by two factors:

1. **Viewpoint sensitivity:** Descriptive views (e.g., the front of a piano) and non-descriptive views (e.g., the back of the same piano) contribute equally to the aggregated embedding. If the majority of views lack discriminative features, the resulting label can shift toward incorrect categories.

2. **Label space constraints:** Even in open-vocabulary settings, relying on a fixed set of candidate labels constrains the level of detail that can be captured for each object, e.g., labeling a chair simply as 'furniture' rather than distinguishing it as an 'office swivel chair.'

To illustrate the severity of this issue, we consider a controlled example where images are taken around a piano. We define **good views** as those from the front, containing distinctive features, and **bad views** as those from the back, lacking such cues. In a progressive experiment, we start with three good views and incrementally replace them with bad ones, testing multiple labeling strategies. The task is to assign a single label to the object, given all current views.

As shown in Table 1, methods relying solely on YOLO or CLIP degrade quickly as bad views increase. In the 1-good / 2-bad case, CLIP-based methods incorrectly label the piano as "crate" or "oak," while YOLO

| Method | 3 Good / 0 Bad | 2 Good / 1 Bad | 1 Good / 2 Bad | |
|---|---|---|---|---|
| YOLOE Constrained | **piano (0.25)** | **piano (0.21)** | crate (0.26) | |
| YOLOE ScanNet200 | cabinet (0.78) | cabinet (0.61) | cabinet (0.51) | |
| YOLOE RAM | chiffonier (0.90) | wall (0.16) | wall (0.17) | |
| CLIP Constrained | **piano (0.31)** | **piano (0.27)** | crate (0.26) | |
| CLIP ScanNet200 | **piano (0.31)** | **piano (0.27)** | crate (0.26) | |
| CLIP RAM | **piano (0.31)** | **piano (0.27)** | oak (0.27) | |
| LLM | **acoustic piano** | **acoustic piano** | **acoustic piano** | |

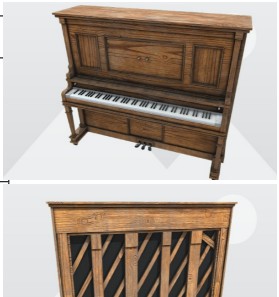

Table 1: **Piano viewpoint bias experiment.** "Good" images show the piano front, while "Bad" images show the back. Each cell reports the *predicted label (confidence)*, with correct predictions shown in **bold**. For YOLOE baselines, the most frequent label is selected, whereas CLIP baselines choose the label with the highest similarity. The "Constrained" setting restricts candidate labels to "piano" and "crate," while "ScanNet200" (Rozenberszki et al., 2022) and "RAM" (Recognize Anything Model class list of over four thousand categories, (Zhang et al., 2023)) use their respective class lists to select the most probable label. The LLM is prompted to give a more specific label than a simple class label."

struggles particularly when the label space is large, producing highly inconsistent predictions. In contrast, the LLM-based approach consistently selects the correct and more detailed label across all conditions. However, the LLM does not provide calibrated confidence values, making it difficult to assess the reliability of its predictions. This observation motivates LADR's hybrid strategy: combining the reasoning capabilities of LLMs with the quantitative confidence scores from CLIP. Our approach allows for both robust label selection and informed weighting across views, mitigating the effects of viewpoint bias and constrained label spaces.

## 4 Notation and Workflow

We consider multi-view object labeling in 3D scenes. For simplicity, and without loss of generality, we address a single object. Let $\mathcal{I} = \{I_1, \ldots, I_N\}$ denote the initial set of RGB-D images captured around a target object, where $N$ is the number of views. Each image $I_i$ is accompanied by depth information $D_i$ and camera pose $P_i$. For each image, object observations are extracted using a combination of a detector, a feature extractor and a segmentation model as

$$O_i = \text{DetectAndSegment}(I_i, D_i, P_i),$$

and merged across all views into a spatial-semantic map

$$\mathcal{M} = \text{MergeObservations}(\{O_1, \ldots, O_N\}),$$

which accumulates object points, labels, and features into a coherent 3D representation, analogous to ConceptFusion's fusion (Jatavallabhula et al., 2023). We define a function for relabeling as

$$\mathcal{M}_{\text{refined}}, \ P_{\text{next}} = \text{RefineAndPropose}(\mathcal{M}, \mathcal{I}),$$

which applies LLM reasoning to refine labels in $\mathcal{M}$ and selects the most informative next viewpoint.

Our workflow proceeds iteratively: images are captured and objects inside it are merged into $\mathcal{M}$, refined, and used to propose the next viewpoint. This repeats until labels reach sufficient confidence or a maximum number of views is obtained.

We note that while DetectAndSegment uses of-the-shelf detector that may fail in detection, and separation of objects, this can be remediated in the RefineAndPropose as the LLM will throw these out, or flip to one of the objects (assuming that these are the sides of the same object) forcing single object for each refinement process.

## 5 Markov process

We formalize the iterative refinement procedure of LADR as a stochastic process over the evolving set of candidate views. At each step, the algorithm samples views, updates a semantic hypothesis via LLM reasoning, and prunes inconsistent observations. This induces a Markovian evolution over a latent state capturing the composition of informative ("good") and misleading ("bad") views. By characterizing the transition dynamics under LLM-guided selection and pruning, we show that the process exhibits a positive drift toward eliminating bad views. Under mild assumptions, this yields exponential convergence to a stable and semantically consistent labeling, providing a theoretical explanation for the robustness observed in practice.

**Markov formulation underlying all algorithms.** Let $I_t$ denote the set of retained views at iteration $t$, and let $h_t$ be the current hypothesis produced by the LLM. We define the full algorithmic state

$$S_t := (I_t, h_t). \tag{1}$$

For each of the three algorithms presented below—LLM-RANDOM, LLM-SAMPLING, and LLM-POLYGON—one iteration consists of the following transition rule:

1. **Selection step.** From the current state $S_t$, compute statistics (random subset in LLM-RANDOM; CLIP-based or polygon-face scores in the other variants) and choose a set of prompt views $\mathcal{P}_t = \mathcal{P}(S_t)$.

2. **LLM update.** Query the LLM with $\mathcal{P}_t$ to obtain a stochastic new hypothesis

$$h_{t+1} \sim \mathcal{L}(\cdot \mid \mathcal{P}_t), \tag{2}$$

   where $\mathcal{L}$ denotes the conditional distribution of the LLM response.

3. **Pruning step.** Using $h_{t+1}$ and deterministic rules specific to the algorithm, update the view set

$$I_{t+1} = \Phi(I_t, h_{t+1}). \tag{3}$$

Since all randomness enters only through equation 2 (and explicit sampling in LLM-RANDOM), the process satisfies the Markov property

$$\Pr(S_{t+1} \in A \mid \mathcal{F}_t) = \Pr(S_{t+1} \in A \mid S_t), \qquad \forall A, \tag{4}$$

where $\mathcal{F}_t = \sigma(S_0, \ldots, S_t)$ is the natural filtration. Hence $\{S_t\}_{t \geq 0}$ is a time-homogeneous Markov chain.

**Coarse processes used in the proofs.** Let $B$ and $G$ denote the sets of bad and good views, and define

$$b_t := |I_t \cap B|, \qquad X_t := \mathbf{1}\{b_t = 0\}. \tag{5}$$

Because $b_t$ and $X_t$ are measurable functions of $S_t$, they are also Markov processes induced by $\{S_t\}$.

Under the algorithm-specific assumptions, whenever $b_t > 0$ there exist constants $\delta > 0$ and $\eta > 1/2$ such that

$$\Pr(b_{t+1} = b_t - 1 \mid \mathcal{F}_t) \geq \beta := \delta\,\eta, \tag{6}$$

representing the probability that the algorithm selects a good–bad pair and the LLM/pruning step eliminates the bad view. Conversely, let

$$\alpha := \sup_t \Pr(b_{t+1} = b_t + 1 \mid \mathcal{F}_t), \tag{7}$$

which upper-bounds the probability of a corruption event (removal of a good view). The gap $\beta - \alpha > 0$ provides the uniform drift toward $b_t = 0$.

From the two-state chain for $X_t$ we obtain

$$p_{t+1} := \Pr[b_{t+1} > 0] \leq \alpha + (1 - \alpha - \beta)\, p_t, \tag{8}$$

which yields exponential decay

$$p_t \leq c_1 e^{-c_2 t}, \qquad c_2 = -\log(1 - \beta + \alpha). \tag{9}$$

This Markov framework—state definition equation 1, transition mechanism equation 2–equation 3, and drift parameters $\alpha, \beta$ in equation 6–equation 7—forms the common basis for the convergence proofs of all three algorithms.

## 6 Methodology

In this section, we present our algorithm for LLM-guided multi-view object labeling. In multi-view labeling, the evolving set of detection images at each iteration often contains a mix of highly informative canonical views, ambiguous and redundant observations. Presenting all available images to the LLM simultaneously is problematic: it risks pushing the model toward a generic, lowest-common-denominator label, substantially increases computational cost, and may even exceed the LLM's context window. A possible workaround is to tile multiple views into a single composite image, but this forces downsampling that discards fine-grained details. To address these challenges, we adopt an iterative inner loop that samples a small subset of images to form a hypothesis and then prunes away views that conflict with it.

We introduce two ablated LADR studies prior to presenting our algorithm, to facilitate the introduction of LADR. We focus on the RefineAndPropose function which defines each algorithm. We also provide convergence analysis for each algorithm, showing improved convergence rates with the addition of key algorithmic components.

### 6.1 LLM-Random: Basic Hypothesis Proposal and Killing

The first ablated version, **LLM-Random**, introduces the fundamental hypothesis-proposal and iterative image removal procedure. The LLM-Random variant implements this process using the simplest possible sampling strategy: uniform random selection. The algorithm pseudo code is given in Alg. 1.

The process repeats until either (1) the LLM reports confidence in its label, or (2) the detection set $\mathcal{I}$ has been reduced to fewer than $N$ images (3) a maximum number of iterations is reached; in which case the algorithm returns the refined map $\mathcal{M}_{\text{refined}}$ along with the next proposed viewpoint $P_{\text{next}}$. The LLM prompt used for this algorithm is provided in Appendix B.7.

#### 6.1.1 Theoretical analysis

For an apples-to-apples comparison with methods that compare two views per iteration, we analyze the randomized baseline in the minimal $N = 2$ setting. The extension to general $N$ follows by replacing pairwise exposure and dominance constants with their $N$-subset counterparts.

**Theorem 1 (High-probability elimination of bad views under LLM-kill random-pair pruning)**
*Let $\mathcal{I}_0$ be a finite set of views, and let $G \subseteq \mathcal{I}_0$ and $B = \mathcal{I}_0 \setminus G$ denote the good and bad views, with $G \neq \varnothing$. At iteration $t$, let $\mathcal{I}_t$ be the retained set and define*

$$G_t := G \cap \mathcal{I}_t, \qquad B_t := B \cap \mathcal{I}_t, \qquad b_t := |B_t|.$$

*Assume the following hold for all $t$:*

**(A1) Exposure under random sampling.** *There exists $\delta > 0$ such that whenever $b_t > 0$,*

$$\Pr(\mathcal{I}_{\text{sample},t} \cap B_t \neq \varnothing \mid \mathcal{F}_t) \geq \delta, \quad \mathcal{F}_t := \sigma(\{X_s : s \leq t\}).$$

---

**Algorithm 1 LLM-Random: Iterative Hypothesis Proposal and View Pruning.** Given a set of detection images and camera poses, the algorithm iteratively samples views and queries a large language model (LLM) to propose a hypothesis $\hat{y}_t$ together with a boolean confidence indicator $c_t$. The confidence indicator denotes whether the LLM considers the current hypothesis sufficiently reliable to terminate the procedure; otherwise, an uninformative view is pruned and the process continues until a confident hypothesis is obtained or a maximum number of iterations is reached.

---

**Require:** Detection images $\mathcal{I}$, camera angles $\mathcal{P}$, sample size $N$
1: $\mathcal{M}_{\text{refined}} \leftarrow \varnothing$ , $t \leftarrow 0$
2: **while** $|\mathcal{I}| \geq N$ **and** $t < T_{\max}$ **do**
3: $\quad t \leftarrow t + 1$
4: $\quad \mathcal{I}_{\text{sample}} \leftarrow \text{RandomSample}(\mathcal{I}, N)$
5: $\quad (\hat{y}, c_t, I_{\text{kill}}, P_{\text{next}}) \leftarrow \text{LLM\_Query}(\mathcal{I}_{\text{sample}}, \mathcal{P})$
6: $\quad$ **if** $c_t = \text{True}$ **then**
7: $\quad\quad \mathcal{M}_{\text{refined}} \leftarrow \hat{y}$
8: $\quad\quad$ **break**
9: $\quad$ **end if**
10: $\quad \mathcal{I} \leftarrow \mathcal{I} \setminus \{I_{\text{kill}}\}$
11: $\quad \mathcal{M}_{\text{refined}} \leftarrow \hat{y}$
12: **end while**
13: **return** $(\mathcal{M}_{\text{refined}}, P_{\text{next}})$

---

**(A2) LLM kill-dominance on mixed pairs.** *There exists $\eta_{\text{kill}} > \frac{1}{2}$ such that whenever $\mathcal{I}_{\text{sample},t} = \{I_g, I_b\}$ with $I_g \in G_t$ and $I_b \in B_t$, the selected image to prune at step $b$ obeys*

$$\Pr(I_{\text{kill},t} = I_b \mid \mathcal{I}_{\text{sample},t} = \{I_g, I_b\}, \ \mathcal{F}_t) \geq \eta_{\text{kill}}.$$

*Define $\beta := \delta \, \eta_{\text{kill}} \in (0, 1)$ and let $b_0 := |B|$.*

$$\text{for any } T \geq 1, \quad \Pr(b_T > 0) \leq \Pr\big(\text{Bin}(T, \beta) < b_0\big). \text{ In particular, if } T \geq \frac{2b_0}{\beta} \text{ then}$$

$$\Pr[b_T > 0] \leq \exp\left(-\frac{\beta T}{8}\right) = \exp(-c_2 T), \qquad c_2 := \frac{\beta}{8}.$$

*Equivalently, for any $\varepsilon \in (0, 1)$ it suffices to take $T \geq \max\left\{\frac{2b_0}{\beta}, \frac{8}{\beta} \log \frac{1}{\varepsilon}\right\}$ to guarantee $\Pr[b_T > 0] \leq \varepsilon$.*

## 6.2 LLM-Sampling: CLIP-Guided Selection and Confidence

The **LLM-Sampling** algorithm follows similar structure as LLM-Random, but improves upon it by leveraging image embeddings provided by a contrastive VLM (eg. CLIP, Cherti et al. (2023)) for both image selection and confidence assessment; we refer to these embeddings as *CLIP features*. We illustrate the method using two sampled images per iteration for clarity. In practice, this generalizes to two subsets of images, sampled similarly. We provide a sketch of an iteration in Figure 1, and pseudo code in Algorightm 2.

**Sampling.** Instead of randomly sampling images, this algorithm identifies two images $I_{\text{rep}}, I_{\text{amb}} \subset \mathcal{I}$ based on their cosine similarity of CLIP features relative to the current label hypothesis: the closest (most representative) and the farthest (potentially ambiguous) image. The initial hypothesis can be set using the most common detection label (e.g. YOLO detections). $I_{\text{rep}}$ and $I_{\text{amb}}$ are then fed to the LLM to generate a new label hypothesis $\mathcal{M}_{\text{refined}}$ and propose the next best view $P_{\text{next}}$. This procedure balances *exploitation* (focusing on the most representative view) with *exploration* (including a diverse, informative view).

**Confidence Computation and Removal.** A global object representation is computed by averaging CLIP features across all current images. Cosine similarity between this global feature and the LLM label embedding $\mathcal{M}_{\text{refined}}$ provides a confidence score for the proposed label. Similarities are computed between

$\hat{y} = \mathcal{M}_{\text{refined}}$ and the sampled detections $I_{rep}, I_{amb}$. The less similar detection is discarded from $\mathcal{I}_t$. To determine whether the current label is reliable enough to be accepted or if further iterations with new images are required, a confidence threshold is applied (see Appendix B for its calibration).

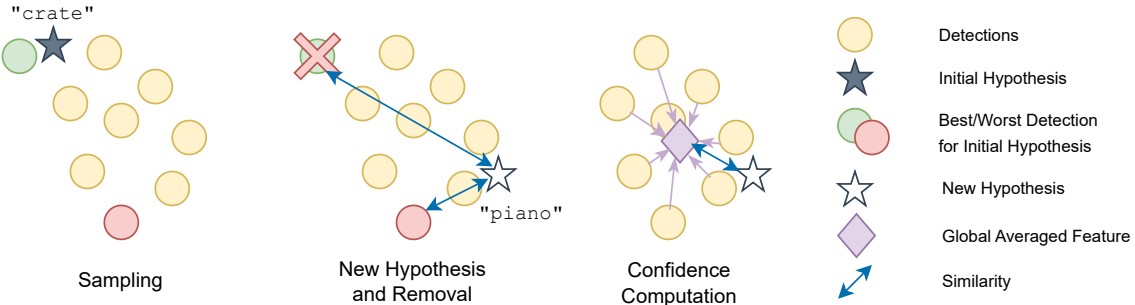

Figure 1: **Visualization of the LLM-Sampling algorithm**: (left) two images are selected based on feature distance from current hypothesis, (middle) a new label hypothesis is generated from the two images, and the less similar detection is removed, (right) a global averaged feature and global confidence are computed.

**Recovering the Final Hypothesis via Caching.** In some cases, the LLM may generate an accurate label early on, but it cannot yet be accepted due to insufficient supporting evidence. The CLIP-similarity-based confidence computation allows for the re-evaluation of previously generated hypotheses. When new images are introduced or when the hypothesis-proposal loop concludes, the most confident hypothesis is retrieved from the cache. Empirically, this mechanism reduces noise from LLM hallucinations, prevents sudden label shifts, and improves convergence consistency.

**Advantages.** By selecting views using CLIP feature distances, our approach presents the LLM with more informative and diverse samples than random subset selection, reducing redundancy and improving sample efficiency. CLIP-based similarity scores also provide a more stable and interpretable confidence signal than the LLM's self-reported confidence used in LLM-Random. In addition, hypothesis caching allows candidate labels to be retained and re-evaluated without repeated LLM calls, improving both efficiency and robustness. Together, these strategies effectively combine the generative strengths of LLMs with the contrastive structure of CLIP models, leading to more reliable and faster convergence.

### 6.2.1 Theoretical analysis

**Theorem 2 (Exponential elimination of bad views under hypothesis-driven extreme selection)**
*Let $\mathcal{I}_0$ be a finite set of views of an object, and let $G \subseteq \mathcal{I}_0$ and $B = \mathcal{I}_0 \setminus G$ denote the sets of* good *and* bad *views, where good views provide reliable evidence of the true canonical label $y^*$ and bad views are ambiguous. Assume $G \neq \varnothing$. At iteration $t$, the algorithm maintains a retained set $\mathcal{I}_t \subseteq \mathcal{I}_0$ and a hypothesis $h_t$.*

*Define:*
$$G_t = G \cap \mathcal{I}_t, \qquad B_t = B \cap \mathcal{I}_t, \qquad b_t = |B_t|.$$

*Assume:*

**(C1) Extreme selection exposes good and bad views.** *There exists $\delta > 0$ such that, whenever $b_t > 0$, For every good–bad pair $\{I_g, I_b\}$ selected as $I_{\text{rep},t}, I_{\text{amb},t}$,*
$$\Pr\big(I_{\text{b},t} \in B_t, \ I_{\text{g},t} \in G_t \mid \mathcal{F}_t\big) \geq \delta, \quad \mathcal{F}_t := \sigma(\{X_s : s \leq t\}).$$

**(C2) Good-vs-bad semantic dominance.** *There exists $\eta > 1/2$ such that, whenever the selected unordered pair $\{I_{\text{rep},t}, I_{\text{amb},t}\}$ consists of exactly one good view $I_g \in G_t$ and one bad view $I_b \in B_t$, the refined hypothesis $h_{t+1}$ produced by the LLM satisfies*
$$\Pr\Big(\cos\big(h_{t+1}, f_{\text{CLIP}}(I_g)\big) > \cos\big(h_{t+1}, f_{\text{CLIP}}(I_b)\big) \Big| \{I_{\text{rep},t}, I_{\text{amb},t}\} = \{I_g, I_b\}, \mathcal{F}_t\Big) \geq \eta.$$

---

**Algorithm 2 LLM-Sampling: Feature-Guided Multi-View Label Refinement.** The algorithm iteratively selects representative and ambiguous views using CLIP features, queries an LLM to propose a label, and prunes inconsistent images based on feature-level agreement, retaining the hypothesis with the highest global consistency score.

---

**Require:** Detection images $\mathcal{I}$, camera angles $\mathcal{P}$, sample size $N = 2$
1: Initialize hypothesis cache $\mathcal{C} \leftarrow \varnothing$
2: $t \leftarrow 0$
3: **while** $|\mathcal{I}| \geq N$ and $t < T_{\max}$ **do**
4:  $t \leftarrow t + 1$
5:  **Compute global representation**: $\mathbf{g} \leftarrow \frac{1}{|\mathcal{I}|} \sum_{I \in \mathcal{I}} f_{\text{CLIP}}(I)$
6:  **Select informative views**: $\forall \mathcal{I}$ : compute $s_I = \cos(f_{\text{CLIP}}(I), \mathbf{g})$
7:   $I_{\text{rep}} \leftarrow \arg\max_{I \in \mathcal{I}} s_I$          ▷ closest to mean, most representative
8:   $I_{\text{amb}} \leftarrow \arg\min_{I \in \mathcal{I}} s_I$          ▷ farthest from mean, most ambiguous
9:   $\mathcal{I}_{\text{sample}} \leftarrow \{I_{\text{rep}}, I_{\text{amb}}\}$
10:  **Query LLM to propose a label**: $(\hat{y}, h, P_{\text{next}}) \leftarrow \text{LLM\_Query}(\mathcal{I}_{\text{sample}}, \mathcal{P})$
11:  **Evaluate hypothesis confidence**: $c_{\text{global}} \leftarrow \cos(h, \mathbf{g})$
12:  **Determine which image harms consistency more**:
13:   **if** $s_{amb} < s_{rep}$ **then**: Remove $I_{\text{amb}}$ from $\mathcal{I}$
14:   **else**: Remove $I_{\text{rep}}$ from $\mathcal{I}$
15:   **end if**
16:  **Cache hypothesis**: Store $(\hat{y}, h, c_{\text{global}})$ in $\mathcal{C}$
17: **end while**
18: **Recover most confident hypothesis**: $(y^*, h^*, c^*) \leftarrow \arg\max_c (\hat{y}, h, c) \in \mathcal{C}$
19: **return** $(y^*, P_{\text{next}})$

---

*Then there exist $c_1, c_2 > 0$ (depending only on $\delta$ and $\eta$) such that for all $T \geq 0$, $\Pr[b_T > 0] \leq c_1 e^{-c_2 T}$.*

*Namely, the probability that at least one bad view remains after $T$ iterations decays exponentially in $T$. Or, for any $\varepsilon \in (0, 1)$, all bad views are eliminated after $O(\log(1/\varepsilon))$ iterations with probability at least $1 - \varepsilon$.*

## 6.3 LLM-Polygon: Spatially Grounded Refinement

The LADR algorithms presentation, **LLM-Polygon**, extends LLM-Sampling by incorporating spatial grounding into the label refinement process. This addition allows the algorithm to reason about coverage of the object's geometry, to guide exploration, and to prioritize views that reduce semantic uncertainty, see Figure 2. We provide the pseudo code for LLM-Polygon in Algorithm 3.

**Spatial Assignment.** A right-prism polygon is constructed around the object to approximate its spatial extent. Each detection is associated with the polygon faces it observes, determined by projecting camera rays on the polygon faces. This partition grounds the detections into spatially meaningful subsets and prevents over-representation of individual sides.

**Per-Face Confidence.** For each polygon face, CLIP features of the associated detections are averaged to form a local feature representation. Unobserved faces are assigned an *uncertainty weight*, a hyperparameter that trades off exploration and exploitation: lower uncertainty weights promote taking additional views, while higher values enable faster convergence by downweighting unseen sides (see Appendix B for a calibration guide). Global confidence is then computed as the average similarity between the current label hypothesis and the per-face features of existing images in $I_t$.

**Iterative Refinement.** Label proposal and image pruning proceed as in LLM-Sampling, but the next viewpoint is chosen using spatial confidence and coverage, and not via an LLM. Specifically, $P_{next}$ is selected as the face whose neighboring faces exhibit the largest confidence difference, with priority given to previously unseen sides. This active mechanism directs exploration toward underrepresented object regions.

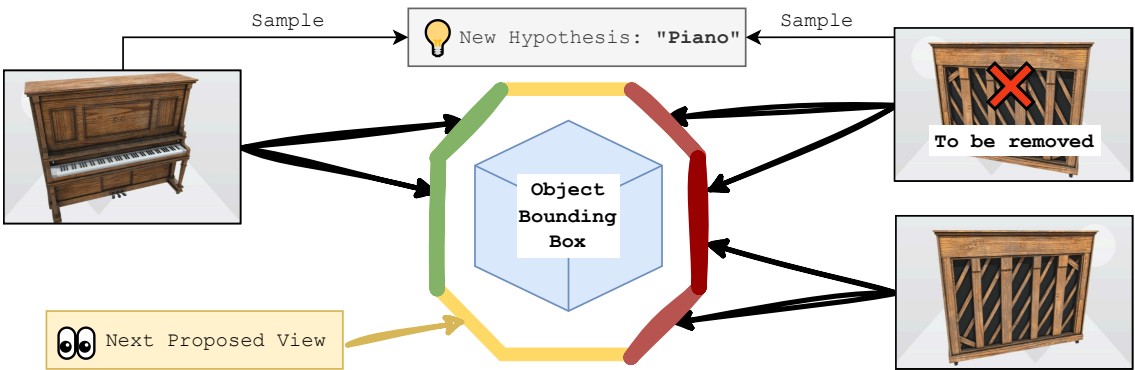

Figure 2: **Illustration of LLM-Polygon**: Object detections are spatially grounded to polygon faces. Per-face confidence is computed based on CLIP features: green sides correspond to high visual similarity to the current label ("piano"), red sides indicate low similarity, and yellow sides represent unseen faces. The next camera viewpoint is selected to reduce uncertainty, prioritizing unseen faces.

### 6.4 Theoretical Analysis

LLM-Polygon introduces three mechanisms absent from LLM-Sampling.

**(A) Spatial partitioning via polygon faces.** Each view is assigned to one or more polygon faces $F \in \mathcal{F}$, based on camera-ray intersection. This ensures balanced geometric coverage and eliminates oversampling of a single side of the object.

**Theoretical implication:** Every side of the object is guaranteed to be observed, so *good views always have nonzero sampling probability*. In LLM-Sampling this had to be assumed; here it is built into the algorithm.

**(B) Per-face feature averaging with uncertainty weighting.** Each face $F$ computes a representative embedding

$$\mathbf{f}_F = \begin{cases} \frac{1}{|\mathcal{I}_F|} \sum\limits_{I \in \mathcal{I}_F} f_{\mathrm{CLIP}}(I), & \mathcal{I}_F \neq \varnothing, \\ \lambda \mathbf{u}, & \mathcal{I}_F = \varnothing, \end{cases}$$

where $\mathbf{u}$ is a calibrated uncertainty vector. Then each image receives a similarity score $s_I = \cos\big(f_{\mathrm{CLIP}}(I), \mathbf{f}_{F(I)}\big)$.

**Theoretical implication:** Bad views tend to be scored poorly (ambiguous), and therefore are much more likely to be selected as $I_{\mathrm{amb}}$. Thus the probability of encountering a good–bad pair increases automatically.

**(C) Spatially grounded viewpoint selection.** The next viewpoint $P_{\mathrm{next}}$ is chosen from faces with large confidence differences $\Delta_F$, prioritizing unseen faces.

**Theoretical implication:** Exploration becomes *systematic and directed*, guaranteeing eventual exposure of faces where inconsistencies (bad views) are likely to reside.

LLM-Polygon is not simply LLM-Sampling with additional notation. Its geometric structure fundamentally changes the sampling distribution, leading to provably stronger constants in the convergence theorem. We now formalize these improvements.

**Theorem 3 (Exponential Decay of Bad Views under LLM-Polygon)** *Let $G_t$ and $B_t$ denote the good and bad views at iteration $t$, and $b_t = |B_t|$. Assume:*

1. *Semantic dominance: For every good–bad pair $\{I_g, I_b\}$ selected as $I_{\mathrm{rep},t}, I_{\mathrm{amb},t}$,*

$$\Pr[\cos(h_{t+1}, f(I_g)) > \cos(h_{t+1}, f(I_b))] \geq \eta > \tfrac{1}{2}.$$

---

**Algorithm 3** LLM-Polygon: Spatially Grounded Multi-View Label Refinement

---

**Require:** Detection images $\mathcal{I}$, camera angles $\mathcal{P}$, polygon faces $\mathcal{F}$, uncertainty weight $\lambda$
1: Initialize hypothesis cache $\mathcal{C} \leftarrow \varnothing$
2: $t \leftarrow 0$
3: **while** $|\mathcal{I}| \geq 2$ and $t < T_{\max}$ **do**
4:     $t \leftarrow t + 1$
5:     **Spatial assignment of detections**:
6:     **for** each face $F \in \mathcal{F}$ **do**
7:         $\mathcal{I}_F \leftarrow$ detections whose camera rays intersect $F$
8:     **end for**
9:     **Compute per-face features and uncertainties**:
10:     **for** each face $F \in \mathcal{F}$ **do**
11:         **if** $\mathcal{I}_F$ is non-empty **then**: $\mathbf{f}_F \leftarrow \frac{1}{|\mathcal{I}_F|} \sum_{I \in \mathcal{I}_F} f_{\mathrm{CLIP}}(I)$
12:         **else**: $\mathbf{f}_F \leftarrow \lambda \cdot \mathbf{u}$                         $\triangleright$ $\mathbf{u}$: calibrated uncertainty vector
13:         **end if**
14:     **end for**
15:     **Select informative views**:
16:     For each detection $I$, assign similarity $s_I = \cos(f_{\mathrm{CLIP}}(I), \mathbf{f}_{F(I)})$
17:     $I_{\mathrm{rep}} \leftarrow \arg\max_{I \in \mathcal{I}} s_I$                   $\triangleright$ closest to mean, most representative
18:     $I_{\mathrm{amb}} \leftarrow \arg\min_{I \in \mathcal{I}} s_I$                  $\triangleright$ farthest from mean, most ambiguous
19:     $\mathcal{I}_{\mathrm{sample}} \leftarrow \{I_{\mathrm{rep}}, I_{\mathrm{amb}}\}$
20:     **Compute spatially grounded global confidence**:
21:     $c_{\mathrm{global}} \leftarrow \frac{1}{|\{F : \mathcal{I}_F \neq \varnothing\}|} \sum_{F : \mathcal{I}_F \neq \varnothing} \cos(h, \mathbf{f}_F)$
22:     **Determine which image harms consistency more**:
23:     **if** $s_{amb} < s_{rep}$ **then**: Remove $I_{\mathrm{amb}}$ from $\mathcal{I}$
24:     **else**: Remove $I_{\mathrm{rep}}$ from $\mathcal{I}$
25:     **end if**
26:     **Cache hypothesis**: Store $(\hat{y}, h, c_{\mathrm{global}})$ in $\mathcal{C}$
27:     **Select next viewpoint via spatial uncertainty**:
28:     For each face $F$, compute $\Delta_F = $ max confidence difference with adjacent faces
29:     $P_{\mathrm{next}} \leftarrow$ viewpoint observing face with largest $\Delta_F$, prioritizing unseen faces
30: **end while**
31: **Recover most confident hypothesis**: $(y^*, h^*, c^*) \leftarrow \arg\max_c (\hat{y}, h, c) \in \mathcal{C}$
32: **return** $(y^*, P_{\mathrm{next}})$

---

    2. ***Geometric exposure of bad views:*** *For every iteration with* $b_t > 0$,

$$\Pr[\{I_{\mathrm{rep},t}, I_{\mathrm{amb},t}\} \cap B_t \neq \varnothing] \geq \delta_{\mathrm{poly}} > 0.$$

    *That is, with probability at least* $\delta_{\mathrm{poly}}$*, the deterministically chosen representative and ambiguous views include at least one bad view. This probability is enforced by polygon-based partitioning, per-face feature averaging, and spatial uncertainty prioritization.*

*Then the probability of retaining any bad views after* $T$ *iterations satisfies* $\Pr[b_T > 0] \leq c_1 e^{-c_2 T}$*, for constants* $c_1, c_2 > 0$ *depending only on* $\eta$ *and* $\delta_{\mathrm{poly}}$*.*

*In particular, the algorithm eliminates all bad views after* $O(\log(1/\varepsilon))$ *iterations with probability at least* $1 - \varepsilon$*.*

**Remark.** The proof follows the same Markov-chain drift argument as for the non-polygon algorithm, with the key constant $\beta$ replaced by $\beta_{\mathrm{poly}} \geq \eta \delta_{\mathrm{poly}}$, where $\delta_{\mathrm{poly}}$ is the probability that the deterministically selected pair $(I_{\mathrm{rep},t}, I_{\mathrm{amb},t})$ forms a good–bad pair. Since polygon-based spatial partitioning and per-face averaging ensure $\delta_{\mathrm{poly}} > \delta_{\mathrm{sample}}$, the drift toward eliminating bad views is strictly larger, yielding a strictly larger exponential rate $c_2$ in the bound

$$\Pr[b_T > 0] \leq c_1 e^{-c_2 T}.$$

### 6.5 Computational Running-Time of LADR Algorithms

In the proposed LADR pipeline, real-time detection is performed by a dedicated detector (e.g., YOLO), while a vision–language model is invoked only for *post-hoc classification* of cropped regions. This design decouples latency-critical localization from semantically rich recognition: YOLO supplies boxes within tens of milliseconds per image, whereas LLaVA or a cloud VLM provides open-vocabulary refinement at the cost of seconds per crop. We provide in Table 2 different options for detection and their wall-clock running time. The table quantifies the trade-off between model size and running time by reporting both end-to-end latency and token-level behavior. Local LLaVA models (7B–13B) achieve sub-second to few-second responses with controllable output lengths, making them suitable for selective verification of ambiguous detections, while API-based models incur additional network overhead. By restricting VLM invocation to a small subset of boxes and capping generated tokens, LADR preserves near-real-time throughput while benefiting from the superior semantic discrimination of large vision–language models.

Table 2: Comparison of detection/classification latency and token behavior. YOLO timing is per image forward pass; LLaVA and ChatGPT timings are per *cropped detection box*. Values reflect typical ranges on modern consumer GPUs with 4–8 bit quantization for local models and short outputs (10–30 tokens).

| Method | Task | Time (GPU) | TTFT | Tokens/s | Output #tok |
|---|---|---|---|---|---|
| YOLO | Box detection (+ class if trained) | 5–20 ms | – | – | – |
| LLaVA 7B (local) | Classify crop | 0.7–2.0 s | 0.15–0.6 s | 20–80 | 10–30 |
| LLaVA 13B (local) | Classify crop | 1.2–3.5 s | 0.25–1.0 s | 12–50 | 10–30 |
| ChatGPT 5.2 API | Classify crop (vision) | 2–8 s | 0.5–3 s | 10–60 | 10–30 |

For local VLM inference the end-to-end latency per crop can be approximated as

$$T_{\text{total}} \approx T_{\text{encode}} + T_{\text{TTFT}} + \frac{N_{\text{out}}}{r_{\text{tok}}},$$

where $T_{\text{encode}}$ is the vision encoder time, $T_{\text{TTFT}}$ is time-to-first-token, $N_{\text{out}}$ is the number of generated tokens, and $r_{\text{tok}}$ is the decoding rate in tokens/s.

## 7 Experiments

We evaluate our proposed method, LADR, against several baseline algorithms in both single-object and multi-object settings. The experiments are designed to assess each method's ability to infer object semantic labels accurately in multi-view scenarios. On single-object scenes, we demonstrate that reasoning with a large language model is crucial for 3D consistency and that active view selection greatly improves sample efficiency and stability. We also evaluate all methods on multi-object scenes, a realistic setting for robot exploration. Here, we isolate the impact of our label generation mechanism by using off-the-shelf exploration policies rather than proposing next-best views. This setup highlights that our representation still improves multi-object labeling as an offline refinement process. Full details of all hyperparameters used are provided in Appendix B.1, and extended results are presented in B.4 and B.5.

### 7.1 Baselines

We compare methods that rely solely on YOLO detections, CLIP embeddings, or LLM reasoning, with LADR, which leverages multi-view aggregation, spatial grounding, and confidence-based label selection.

**'YOLO'**: uses the most common YOLO label as the final label from a closed vocabulary. This is the aggregation policy in ConceptFusion (Jatavallabhula et al., 2023).

**'CLIP'**: takes the average of the CLIP embeddings of all images and compares it via cosine similarity to an extensive list of CLIP-embedded labels of the RAM class list (Zhang et al., 2023). Closed-vocabulary list.

**'LLM-Label'**: The LLM reasons over the set of YOLO labels, their frequencies, and possible semantic relationships to infer the most plausible label. No visual data is used, only text.

**'LLM-Tiled'**: creates a single image with all input images tiled. This layout is then analyzed by a large vision model to produce the final label. See example in Appendix B.6.

**'LLM-Angle'**: creates a single composite image with all input images around a circle capturing their relative positions to the object. This panoramic-style layout is then analyzed by a large vision model to produce the final label. Unlike the other baselines, the LLM provides the next best view to take an image from. See example in Appendix B.6.

**LADR implementation**: Here, LADR refers to our three algorithms: LLM-Random, LLM-Sampling, and LLM-Polygon. Apart from these, only 'LLM-Angle' explicitly proposes the next best view; for all other baselines, random view sampling is used when not otherwise specified.

## 7.2 Datasets

We evaluate our methods on a comprehensive suite of datasets, including a simulated single-object dataset derived from OmniObjects3D, a custom-generated simulated multi-object dataset, and the real-world ARKitScenes dataset.

The data set *OmniObjects3D* (Wu et al., 2023) is a dataset of annotated 3D object models. These objects are rendered in NVIDIA Isaac Sim under controlled conditions to generate multi-view image sequences. We focus on five object classes: backpack, cup, cabinet, sofa, and suitcase. For each class, we include five distinct instances, several of which are deliberately misleading in appearance (e.g., a mug shaped like a cartoon character) to test the robustness of semantic labeling methods.

We also constructed a multi-object dataset in the same simulation environment. These scenes contain multiple objects arranged in varied environments, including SimpleRoom, Commercial, Industrial, Residential, and Vegetation, providing more complex scenarios with occlusions.

Each object in the datasets is annotated with both its class name and a concise descriptive phrase, for example a chair labeled as *chair* with the description *wooden dining chair with a cushioned seat*. We provide examples for both datasets in Appendix B.3.

Finally we provide a real-life data sets study from Baruch et al. (2021) in which we use the baseline to asses their coping with real life challenges including severe occlusion, challenging background, limited collection of view points and more.

## 7.3 Evaluation Metrics

To assess performance, the predicted labels are compared against ground-truth object class names and longer, descriptive phrases (e.g., "yellow cartoon character-shaped mug"). Since LADR is an open-vocabulary setting, direct comparison with ground-truth labels is not sufficient: the LLM may propose synonyms of the annotated class, which should be accepted. Empirically, we found that the CLIP model used for image–text similarity is overly sensitive to lexical variation (e.g., number of words), leading to unreliable synonym matching. Instead, we employ a Sentence Transformer (Reimers & Gurevych, 2019) model to evaluate label equivalence. The final similarity score is defined as the maximum of the similarity to the class name and the similarity to the description, capturing both category-level and instance-level alignment. To evaluate success rates rather than raw similarities, we adopt the similarity value 0.5 as the threshold for label correctness (based on preliminary experiments; see Appendix B.2), while also considering thresholds of 0.3, 0.7, and 0.9.

To evaluate detections in the multi-object setting, we establish one-to-one matches between ground-truth objects and predicted detections from the global map. Matching is based on a semantic-spatial similarity score, defined as a weighted sum of label similarity and spatial overlap between ground-truth and predicted bounding boxes. Once matches are established, evaluation metrics follow the same procedure as in the single-object setting, ensuring comparability.

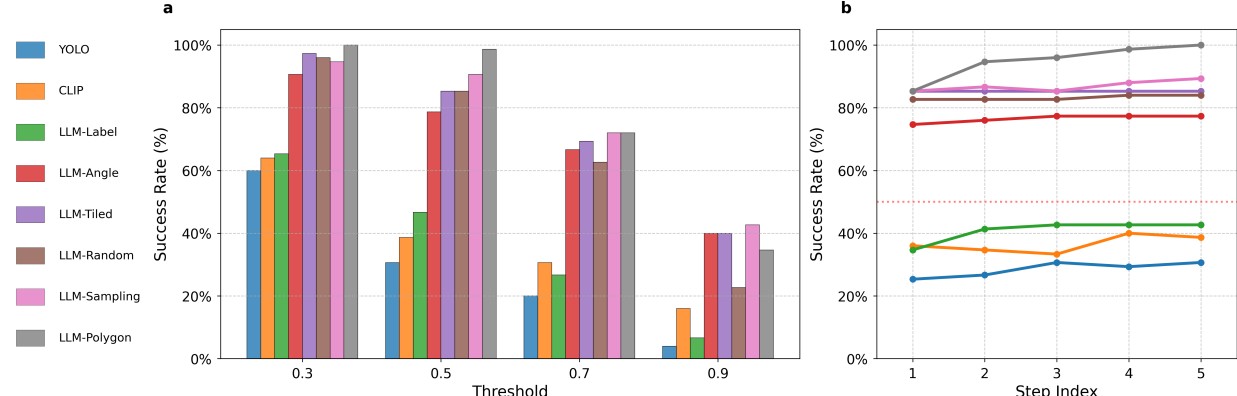

Figure 3: **Single-Object Experiment Results** (a) Averaged success rates across different success thresholds for each algorithm. (b) Evolution of success rates over data collection steps for each algorithm, using 0.5 as the threshold.

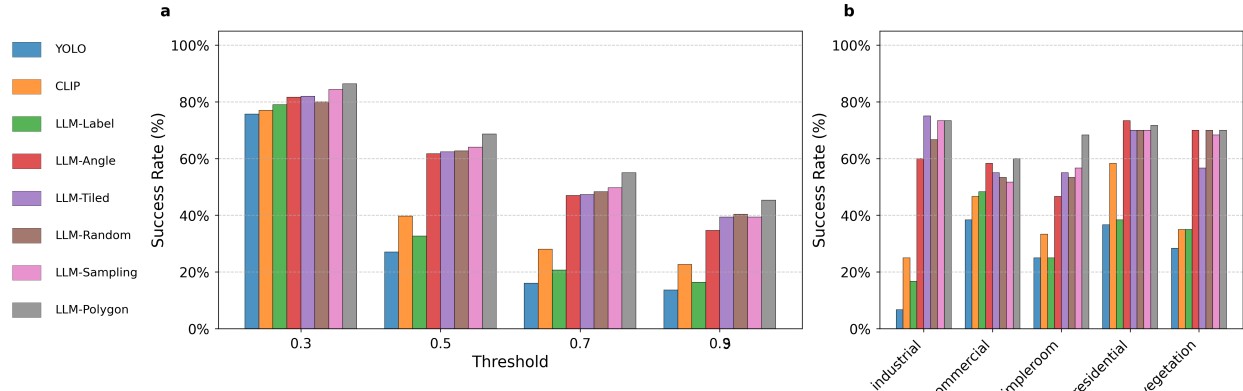

Figure 4: **Multi-Object Experiment Results** (a) Averaged success rates across different success thresholds. (b) Averaged success rates across scenes, using 0.5 as the threshold.

## 7.4 Summary of Findings

We provide our results for the single- and multi-object cases in Figures 3 and 4, respectively. We provide detailed results, including per-object examples for each setting in Appendices B.4 and B.5. Figure 3a shows the averaged success rates based on different success thresholds, and Figure 3b shows how success rates evolve over the data collection steps with 0.5 as the threshold.

Similar trends are observed for single- and multi-object cases. The first observation is that they show over 40% improvements, respectively, compared to ubiquitous fusion methods using YOLO, and CLIP. **YOLO** and **LLM-Label** rely solely on YOLOE predictions, resulting in consistently low success rates. This is likely due to their lack of multi-view image-based reasoning. Notably, LLM reasoning alone offers little improvement over simply taking the most frequent YOLO label. **CLIP** performs comparably to YOLO, but struggles with the vast label set and ambiguity introduced by averaging embedded views, often leading to confused predictions. **LLM-Tiled** achieves higher success rates by leveraging all views simultaneously. However, its accuracy lags behind LADRs, suggesting that the tiled representation loses fine-grained detail or introduces structural incoherence that limits reasoning. **LLM-Angle** adds structural consistency by ordering views in a layout, yet provides no improvement over LLM-Tiled. This indicates that the performance gap is more likely due to loss of visual detail. **LLM-Random** and **LLM-Sampling** analyze images in greater detail, leading to stronger descriptive accuracy. However, LLM-Random often declares detections prematurely, and LLM-Sampling cannot fully mitigate instability despite its confidence-based pruning. Finally, **LLM-Polygon** outperforms all, with near-perfect success at a 0.5 threshold. By combining detailed reasoning with active

exploration and consistency across unseen sides, it avoids the pitfalls of LLM-Only and LLM-Sampling. Figure 3 /b shows how active exploration of unseen sides leads to success rate improvement. LADR's combination of uncertainty sampling, confidence computation, and spatial grounding is key to outperform approaches that provide multiple images to an LLM, as in LLM-Tile and LLM-Angle.

### 7.5 Real-World Data Study

While our primary evaluation is conducted on controlled synthetic and multi-object environments, real-world RGB-D data presents additional challenges that are not fully captured by existing open-vocabulary 3D benchmarks. In particular, real-world datasets provide cluttered, occluded indoor scenes, dense 3D geometry, and often limited or no ability to collect additional views.

We use scene 4733462 from ARKit Scenes Baruch et al. (2021) for our study. While our quantitative evaluation hinges on semantic similarity between ground-truth labels and open-vocabulary predictions. On real-world datasets such as ARKit Scenes, ground-truth labels are limited to a small, coarse taxonomy (e.g., "cabinet", "chair", "table"), while our approach often produces more specific labels (e.g., "armoire", "computer chair", "flat-screen television"). Direct similarity-based scoring in this setting becomes unreliable, as lexical variation and differences in granularity can dominate semantic correctness. For these reasons, instead of reporting potentially misleading quantitative scores, we present a qualitative case study on ARKit Scenes to analyze algorithmic behavior under realistic sensor noise and clutter. We note that in this case there is no ability to capture every view – as we are limited to a particular video sequence. Nevertheless, the study shows better capability of our method to capture and label correctly the objects, only from views provided in the video sequence.

**Challenges in our Real-World data set.** This scene exhibits several real-world difficulties:

- Heavy occlusions (cabinet partially visible behind door)

- Textureless surfaces (door vs cabinet ambiguity)

- Partial crops dominated by small objects (desk vs mouse pad)

- Missed detections (tv monitor)

- Overlapping bounding boxes

- Background distractors

These phenomena mirror failure modes present in our synthetic benchmarks, but occur here under real capture and reconstruction artifacts, offering a complementary stress-test of semantic consistency.

**Observations**. We provide Table 3 for our current distinct object detection and classification results:

**YOLO (constrained labels)** correctly assigns coarse labels but lacks descriptive specificity [bed vs bean-bag] (and may be sensitive to overlap ambiguities).
**CLIP-aggregation** produces semantically plausible but unstable labels, including occasional hallucinations (e.g., "balloon" under occlusion, [cabinet detection thrown off]), reflecting sensitivity to averaged feature embeddings.
**LLM-Label** improves stability by filtering inconsistent labels but struggles with heavily occluded objects [cabinet occluded behind door].
**LLM-Tiled** produce detailed and generally accurate labels but remain vulnerable to ambiguity when fine-grained details are lost in composite representations [door vs cabinet].
**LLM-Random** yields highly descriptive labels but may overfit to dominant visual captures, such as predicting "extended mouse pad" instead of "computer desk."
**LLM-Sampling** demonstrates targeted disambiguation by iteratively suppressing inconsistent views [cabinet vs door occlusion, balloon, mouse pad on table]. In ambiguous cabinet-versus-door cases, the method selectively removes misleading door-dominant views and converges to the correct interpretation. Labels remain detailed while avoiding overconfident hallucinations. Reported confidences show strong correlation

| GT | YOLO-const. | CLIP (conf) | LLM-Label | LLM-Tiled | LLM-Random | LLM-Sampling | LLM-Polygon | Notes |
|---|---|---|---|---|---|---|---|---|
|  cabinet | cabinet (4 detections) | locker (0.28) | door | wooden interior door | wooden wardrobe | locker (3 q, 1 k, 0.29) | locker (3 q, 1 k, 0.26) | Low number of detections, heavy occlusion by door |
|  bed | bed (23 detections) | bed (0.33) | bed | single metal bed | single metal-frame bed | bed (0 q, 0.33) | bed (0 q, 0.32) | Many detections, strong visual cues, all converge |
|  cabinet | cabinet (7 detections) | balloon (0.30) | wardrobe | wardrobe door | wooden storage cabinet | armoire (3 q, 2 k, 0.30) | wardrobe (3 q, 2 k, 0.29) | Severe feature drift under occlusion; reasoning suppresses bias. |
|  cabinet | cabinet (13 detections) | drawer (0.32) | dresser drawer | dresser | dresser | drawer (0 q, 0.32) | drawer (1 q, 1 k, 0.32) | Granularity-sensitive disambiguation |
|  table | table (5 detections) | computer desk (0.29) | computer desk | computer desk | extended mouse pad | computer desk (2 q, 1 k, 0.32) | wooden desk (3 q, 1 k, 0.31) | Low number of detections; LLM-Random fooled by crops. |
|  tv_monitor | tv_monitor (13 detections) | television (0.29) | screen | flat-screen television | flat-screen television | flat-screen (1 q, 0 k, 0.32) | flat-screen TV (2 q, 1 k, 0.33) | Stable across strategies. Some detections show only screen. |
|  chair | bed (8 detections) | beanbag (0.31) | beanbag | round bean bag pouf | bean bag chair | beanbag (1 q, 1 k, 0.33) | beanbag (1 q, 1 k, 0.32) | Open-vocabulary refinement increases specificity. |
|  chair | chair (19 detections) | computer chair (0.34) | computer chair | office chair | office/gaming chair | computer chair (0 q, 0.34) | computer chair (0 q, 0.33) | Consistent semantic refinement with preserved correctness. |

Table 3: Detection results with ground truth (GT) images and class labels. Confidences are rounded to 2 decimals. Polygon has slightly lower confidences, due to the unseen sides downweighting the average. The table illustrates algorithmic behavior rather than quantitative superiority, as ARKit provides only coarse semantic labels unsuitable for open-vocabulary similarity scoring.

with actual ambiguity of detections.

**LLM-Polygon** exhibits similar behavior to LLM-Sampling in this scene, though the spatial reasoning component is less critical due to limited ability to capture every view.

# 8 Conclusion

This paper introduces LLM-guided Active Detection and Reasoning (LADR), a novel open-vocabulary algorithm that significantly advances 3D-consistent visual reasoning by leveraging large language models (LLMs) to mitigate challenges in multi-view semantic object detection.

## 8.1 Key Conclusions

- **Addressing 3D Consistency:** LADR effectively resolves the persistent problem of inconsistent semantic labels across multiple viewpoints, which often arises from non-descriptive views, occlusions, and the limitations of fixed label lists in existing zero-shot approaches.

- **LLM-Guided Relabeling:** The core innovation lies in using LLM reasoning to actively refine and reweight multi-view detections. This process corrects viewpoint-induced misclassifications without requiring task-specific retraining and eliminates the need for a predefined label set, enabling truly open-vocabulary capabilities.

- **Smart Sampling Strategy:** LADR incorporates an intelligent active sampling mechanism that selects informative views based on feature diversity, uncertainty estimation, and geometric grounding. This balances detail preservation with reduced context ambiguity, leading to more efficient and robust label refinement.

- **Spatial-Semantic Mapping:** The method constructs a robust spatial-semantic representation by integrating refined labels with object geometry, making it suitable for various downstream 3D tasks.

- **Superior Performance:** Comprehensive evaluations on controlled single-object and multi-object scenes demonstrate substantial improvements, achieving double-digit gains in accuracy and sampling rate compared to ubiquitous fusion methods utilizing YOLO, CLIP, and other LLM-based baselines.

- **Real-World Robustness:** A real-world dataset study confirms LADR's effectiveness in challenging scenarios, including severe occlusion, complex backgrounds, and limited viewpoints.

- **Theoretical Foundation:** The paper provides a rigorous theoretical analysis, formalizing LADR's iterative refinement as a Markov process and proving exponential convergence to a correct and stable semantic label, with stronger constants for enhanced algorithmic components like LLM-Sampling and LLM-Polygon.

In summary, LADR establishes a robust framework for zero-shot open-vocabulary 3D understanding, combining semantic reasoning, efficient view selection, and spatial integration to achieve more consistent and reliable object labeling across diverse and challenging environments.

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

# A  Appendix

## A.1  Proof of Theorem 1

**Step 1: A per-iteration lower bound on removing a bad view.**  *Fix $t$ with $b_t > 0$. Let $E_t$ be the event that the sampled pair contains at least one bad view:*

$$E_t := \{\mathcal{I}_{\text{sample},t} \cap B_t \neq \varnothing\}.$$

*By (A1), $\Pr(E_t \mid \mathcal{F}_t) \geq \delta$. On $E_t$, either the sampled pair is mixed (one good, one bad) or bad–bad. If it is mixed, (A2) gives that the LLM kills the bad view with probability at least $\eta_{\text{kill}}$. If it is bad–bad, the LLM necessarily kills a bad view with probability 1. Therefore,*

$$\Pr(b_{t+1} = b_t - 1 \mid E_t, \mathcal{F}_t) \ \geq \ \eta_{\text{kill}}.$$

*Combining with $\Pr(E_t \mid \mathcal{F}_t) \geq \delta$ yields*

$$\Pr(b_{t+1} = b_t - 1 \mid \mathcal{F}_t) \ \geq \ \delta\,\eta_{\text{kill}} \ := \ \beta, \qquad \text{whenever } b_t > 0. \tag{10}$$

*Define the indicator of a "bad-removal success" at time $t$:*

$$Y_t := \mathbf{1}\{b_{t+1} = b_t - 1\}.$$

*By equation 10, we have for all $t$ with $b_t > 0$,*

$$\Pr(Y_t = 1 \mid \mathcal{F}_t) \ \geq \ \beta.$$

*Let $S_T := \sum_{t=0}^{T-1} Y_t$ be the total number of bad views removed by time $T$. Since $b$ decreases by exactly one when $Y_t = 1$ and never increases,*

$$b_T \ = \ b_0 - S_T \quad \text{as long as } S_T \leq b_0,$$

*and in particular,*

$$\{b_T > 0\} \ \subseteq \ \{S_T < b_0\}.$$

*A standard coupling (or stochastic domination) argument implies that $S_T$ stochastically dominates a binomial random variable with parameters $(T, \beta)$:*

$$S_T \ \succeq \ \text{Bin}(T, \beta),$$

*hence*

$$\Pr(b_T > 0) \ \leq \ \Pr(S_T < b_0) \ \leq \ \Pr(\text{Bin}(T, \beta) < b_0).$$

*Assume $T \geq \frac{2b_0}{\beta}$ so that $\mathbb{E}[\text{Bin}(T, \beta)] = \beta T \geq 2b_0$. Then $b_0 \leq \frac{1}{2}\beta T$, and a Chernoff bound gives*

$$\Pr(\text{Bin}(T, \beta) < b_0) \ \leq \ \Pr\big(\text{Bin}(T, \beta) \leq \tfrac{1}{2}\beta T\big) \ \leq \ \exp\left(-\frac{\beta T}{8}\right).$$

*Therefore,*

$$\Pr(b_T > 0) \ \leq \ \exp\left(-\frac{\beta T}{8}\right),$$

*which is an exponential tail in $T$ with rate $c_2 = \beta/8$. Finally, to make $\Pr(b_T > 0) \leq \varepsilon$, it suffices that $\exp(-\beta T/8) \leq \varepsilon$, i.e. $T \geq \frac{8}{\beta}\log(1/\varepsilon)$, together with $T \geq \frac{2b_0}{\beta}$ to ensure we are in the regime above.*

## A.2  Proof of Theorem 2 - Convergence of LLM sampling Algorithm

**Proof 1 Proof 2** *We show that under (C1)–(C2), the number of bad views $b_t$ decreases with a uniform positive drift, which yields exponential decay of $\Pr[b_T > 0]$.*

**State and evolution.** *The retained set $\mathcal{I}_t$ evolves by removing exactly one view per iteration. Thus*

$$|\mathcal{I}_{t+1}| = |\mathcal{I}_t| - 1, \qquad b_{t+1} \in \{b_t, b_t - 1\} \quad \text{for all } t.$$

*All randomness arises from the LLM's internal sampling when producing $h_{t+1}$ (given the current pair and context). Let $S_t$ denote the full state at time $t$ (including $\mathcal{I}_t$ and $h_t$), and let $\mathcal{F}_t$ denote the sigma-algebra generated by $\{S_0, \ldots, S_t\}$.*

*Conditioned on $S_t$, the selection of $I_{\text{rep},t}$ and $I_{\text{amb},t}$, and the subsequent removal rule, are fully deterministic except for the random LLM response. Since the latter depends only on the current query (which is itself a function of $S_t$), the process $\{S_t\}$ is a Markov chain. Consequently, the derived process $\{b_t\}$ is also Markov.*

**Lower bound on the probability of removing a bad view.** *Fix $t$ with $b_t > 0$, and consider the probability of removing a bad view at iteration $t$.*

*By (C1), conditional on the history $\mathcal{F}_t$, with probability at least $\delta$ we have*

$$I_{\text{amb},t} \in B_t \quad \text{and} \quad I_{\text{rep},t} \in G_t.$$

*On that event, the unordered pair $\{I_{\text{rep},t}, I_{\text{amb},t}\}$ consists of exactly one good and one bad view, say $\{I_g, I_b\}$ with $I_g \in G_t$ and $I_b \in B_t$.*

*The algorithm then queries the LLM, obtains $h_{t+1}$, and removes the view with smaller similarity to $h_{t+1}$ in CLIP space. It removes the bad view $I_b$ if and only if*

$$\cos\big(h_{t+1}, f_{\text{CLIP}}(I_b)\big) < \cos\big(h_{t+1}, f_{\text{CLIP}}(I_g)\big).$$

*By (C2),*

$$\Pr\Big(\cos(h_{t+1}, f(I_g)) > \cos(h_{t+1}, f(I_b)) \,\Big|\, \{I_{\text{rep},t}, I_{\text{amb},t}\} = \{I_g, I_b\}, \mathcal{F}_t\Big) \geq \eta.$$

*Combining these two facts, we obtain*

$$\begin{aligned}
\Pr\big(b_{t+1} = b_t - 1 \mid \mathcal{F}_t, b_t > 0\big) &= \Pr\big(a \text{ bad view is removed at step } t \mid \mathcal{F}_t, b_t > 0\big) \\
&\geq \Pr\big(I_{\text{amb},t} \in B_t, \ I_{\text{rep},t} \in G_t \mid \mathcal{F}_t, b_t > 0\big) \cdot \Pr\big(bad \text{ is removed} \mid good\text{–}bad \text{ pair}, \mathcal{F}_t\big) \\
&\geq \delta \cdot \eta.
\end{aligned}$$

*Define*

$$\beta := \delta\eta > 0.$$

*Then, whenever $b_t > 0$,*

$$\Pr\big(b_{t+1} = b_t - 1 \mid \mathcal{F}_t\big) \geq \beta. \tag{11}$$

**Upper bound on the probability of removing a good view.** *Similarly, we can upper bound the probability that a good view is removed at iteration $t$ while bad views remain.*

*When the selected pair is good–bad, (C2) implies that the good view is removed with probability at most $1 - \eta < 1/2$. When the pair is good–good or bad–bad, removing a good view can only happen in the good–good case. Aggregating these possibilities, we can bound*

$$\Pr\big(a \text{ good view is removed at step } t \mid \mathcal{F}_t, b_t > 0\big) \leq \alpha$$

*for some constant $\alpha < \beta$. (If the algorithm never removes a good view as long as bad views remain, we may take $\alpha = 0$; in general a loose but finite upper bound suffices.)*

**Reduction to a two-state chain.** *Define the binary indicator*

$$X_t = \mathbf{1}\{b_t = 0\} = \begin{cases} 1, & b_t = 0 \quad \textit{(no bad views remain)}, \\ 0, & b_t > 0 \quad \textit{(at least one bad view remains)}. \end{cases}$$

*Then $X_t$ is a Markov chain on $\{0, 1\}$.*

*From equation 11, when $X_t = 0$ (i.e., $b_t > 0$), we have*

$$\Pr(X_{t+1} = 1 \mid X_t = 0) = \Pr(b_{t+1} = 0 \mid b_t > 0) \geq \beta,$$

*so*

$$\Pr(X_{t+1} = 0 \mid X_t = 0) \leq 1 - \beta.$$

*When $X_t = 1$ (no bad views remain), the probability of transitioning to $X_{t+1} = 0$ is at most $\alpha$ by the above argument (interpreting any future reintroduction of bad views as a "corruption" event), so*

$$\Pr(X_{t+1} = 0 \mid X_t = 1) \leq \alpha, \qquad \Pr(X_{t+1} = 1 \mid X_t = 1) \geq 1 - \alpha.$$

*Let $p_t = \Pr[X_t = 0] = \Pr[b_t > 0]$ be the probability that at least one bad view remains at time $t$. Then*

$$\begin{aligned} p_{t+1} &= \Pr[X_{t+1} = 0] \\ &= \Pr(X_{t+1} = 0 \mid X_t = 0)\Pr(X_t = 0) + \Pr(X_{t+1} = 0 \mid X_t = 1)\Pr(X_t = 1) \\ &\leq (1 - \beta)p_t + \alpha(1 - p_t) \\ &= \alpha + (1 - \alpha - \beta)p_t. \end{aligned}$$

*Let*

$$q := 1 - \alpha - \beta.$$

*Since $0 \leq \alpha < \beta \leq 1$, we have $-1 < q < 1$, hence $|q| < 1$. The recurrence becomes*

$$p_{t+1} \leq \alpha + q p_t.$$

**Exponential decay.** *The linear recurrence*

$$p_{t+1} \leq \alpha + q p_t, \quad |q| < 1,$$

*has the general solution*

$$p_t \leq p^* + (p_0 - p^*)q^t, \qquad p^* := \frac{\alpha}{\alpha + \beta}.$$

*Because $|q| < 1$, there exists $c_2 > 0$ such that*

$$|q|^t = e^{t \ln |q|} = e^{-c_2 t}, \quad c_2 := -\ln |q| > 0,$$

*and therefore*

$$p_t \leq p^* + |p_0 - p^*| e^{-c_2 t} \leq c_1 e^{-c_2 t}$$

*for some constant $c_1 > 0$ (absorbing $p^*$ and $|p_0 - p^*|$).*

*Thus*

$$\Pr[b_T > 0] = p_T \leq c_1 e^{-c_2 T},$$

*which proves the first claim.*

**Logarithmic sample complexity.** *Given $\varepsilon \in (0, 1)$, choose*

$$T(\varepsilon) = \left\lceil \frac{1}{c_2} \log \frac{c_1}{\varepsilon} \right\rceil = O(\log(1/\varepsilon)).$$

*Then for all $T \geq T(\varepsilon)$, $\Pr[b_T > 0] \leq \varepsilon$, i.e., with probability at least $1 - \varepsilon$ all bad views have been removed by time $T(\varepsilon)$. This completes the proof.*

### A.3   Why a Markov Chain Model and Why Exponential Decay?

In this section we explain why it is natural (and essentially necessary) to model our refinement procedure as a Markov chain, and how the exponential error decay arises from this viewpoint.

#### A.3.1   Markov structure of the refinement process

Recall that at iteration $t$ the algorithm maintains a retained set of views $\mathcal{I}_t \subseteq \mathcal{I}_0$ and a current hypothesis $h_t$ output by the LLM. We denote the full algorithmic state by

$$S_t = (\mathcal{I}_t, h_t).$$

The transition from $S_t$ to $S_{t+1}$ proceeds as follows:

1. For every $I \in \mathcal{I}_t$, compute the similarity

$$s_I^{(t)} = \cos\big(f_{\mathrm{CLIP}}(I), h_t\big).$$

2. Select the *closest* and *farthest* views with respect to $h_t$:

$$I_{\mathrm{rep},t} = \arg\max_{I \in \mathcal{I}_t} s_I^{(t)}, \qquad I_{\mathrm{amb},t} = \arg\min_{I \in \mathcal{I}_t} s_I^{(t)}.$$

3. The ordered pair $(I_{\mathrm{rep},t}, I_{\mathrm{amb},t})$ and any auxiliary context (e.g., poses) are passed to the LLM, which stochastically produces a new hypothesis $h_{t+1}$.

4. The algorithm compares the similarity of $h_{t+1}$ to the CLIP embeddings of $I_{\mathrm{rep},t}$ and $I_{\mathrm{amb},t}$, and deterministically removes the less supported view, yielding $\mathcal{I}_{t+1}$.

Crucially, *given the current state* $S_t$, the distribution of $S_{t+1}$ depends only on $S_t$ and not on earlier states $S_{t-1}, S_{t-2}, \dots$. The selection of $I_{\mathrm{rep},t}, I_{\mathrm{amb},t}$ is deterministic given $S_t$; all randomness in the transition comes from the internal stochasticity of the LLM when producing $h_{t+1}$, which is conditionally independent of the past given the current query.

Formally, if $\mathcal{F}_t$ is the sigma-algebra generated by $\{S_0, \dots, S_t\}$, then for any measurable set $A$ in the state space,

$$\Pr(S_{t+1} \in A \mid \mathcal{F}_t) = \Pr(S_{t+1} \in A \mid S_t).$$

Thus $\{S_t\}_{t \geq 0}$ is a (time-inhomogeneous) Markov chain.

Since we are interested only in whether any bad views remain, we consider the coarse-grained process

$$b_t = |B_t|, \quad B_t = B \cap \mathcal{I}_t,$$

or equivalently the binary process

$$X_t = \mathbf{1}\{b_t = 0\} = \begin{cases} 1, & \text{if no bad views remain at time } t, \\ 0, & \text{if at least one bad view remains.} \end{cases}$$

Because $X_t$ is a measurable function of $S_t$, the sequence $\{X_t\}_{t \geq 0}$ is also a Markov chain on the state space $\{0, 1\}$. Studying the long-run behavior of the algorithm therefore reduces to analyzing this two-state Markov chain.

The Markov modelling is essential here: the algorithm is an *iterative, stochastic* procedure, and any statement about its asymptotic behavior (e.g., "the probability of retaining bad views goes to zero") must be formulated in terms of the evolution of the distribution of states. The Markov property gives us a clean dynamical equation for this evolution and allows us to derive explicit rates.

### A.3.2 Deriving exponential decay from the Markov chain

Under the assumptions of Theorem 2, we have the following two key properties whenever $B_t \neq \varnothing$:

- With probability at least $\delta > 0$ (condition (C1)), the extreme selection step identifies a good–bad pair, i.e.
$$I_{\text{rep},t} \in G_t \quad \text{and} \quad I_{\text{amb},t} \in B_t.$$

- Conditioned on this event, with probability at least $\eta > 1/2$ (condition (C2)) the LLM-based refinement step removes the bad view rather than the good one (good-vs-bad semantic dominance).

Combining these, we obtain a uniform lower bound on the probability of removing a bad view at iteration $t$:
$$\Pr\big(b_{t+1} = b_t - 1 \,\big|\, \mathcal{F}_t,\, b_t > 0\big) \;\geq\; \beta := \delta\eta > 0.$$

By a symmetric (though weaker) argument, we can upper bound the probability of removing a good view when both good and bad views are present by some constant $\alpha < \beta$. Taken together, these bounds imply that the binary process $X_t = \mathbf{1}\{b_t = 0\}$ satisfies
$$\Pr(X_{t+1} = 1 \mid X_t = 0) \;\geq\; \beta, \qquad \Pr(X_{t+1} = 0 \mid X_t = 1) \;\leq\; \alpha$$

for all $t$, with $0 \leq \alpha < \beta \leq 1$. Thus $\{X_t\}$ is a two-state Markov chain with transition probabilities bounded by
$$P = \begin{pmatrix} P(0 \to 0) & P(0 \to 1) \\ P(1 \to 0) & P(1 \to 1) \end{pmatrix} \approx \begin{pmatrix} 1 - \beta & \beta \\ \alpha & 1 - \alpha \end{pmatrix}.$$

Let $p_t = \Pr[X_t = 0]$ be the probability that at least one bad view remains at time $t$. Using the above bounds, we obtain the linear recurrence
$$\begin{aligned} p_{t+1} &= \Pr[X_{t+1} = 0] \\ &= \Pr[X_{t+1} = 0 \mid X_t = 0]\Pr[X_t = 0] + \Pr[X_{t+1} = 0 \mid X_t = 1]\Pr[X_t = 1] \\ &\leq (1 - \beta)p_t + \alpha(1 - p_t) \\ &= \alpha + (1 - \alpha - \beta)p_t. \end{aligned}$$

Define
$$q := 1 - \alpha - \beta.$$

Since $\alpha < \beta$, we have $-1 < q < 1$ and hence $|q| < 1$. The recurrence can be written as
$$p_{t+1} \leq \alpha + q p_t.$$

Iterating this inequality yields
$$p_t \;\leq\; p^* + (p_0 - p^*)q^t, \qquad p^* := \frac{\alpha}{\alpha + \beta}.$$

Because $|q| < 1$, there exists $c_2 > 0$ such that
$$|q|^t = e^{t \ln |q|} = e^{-c_2 t}, \qquad c_2 := -\ln|q| > 0,$$

and therefore
$$p_t \;\leq\; p^* + |p_0 - p^*|\, e^{-c_2 t} \;\leq\; c_1 e^{-c_2 t}$$

for a suitable constant $c_1 > 0$ (absorbing $p^*$ into the constant if desired).

In other words, the probability that the algorithm has not yet eliminated all bad views by time $t$ decays *exponentially* in $t$, with rate governed by the gap $\beta - \alpha$ between the probability of correcting and the probability of corrupting. This exponential behavior is a direct consequence of viewing the refinement procedure as a Markov chain and analyzing the resulting linear recurrence for the error probability.

### A.4 proof of Theorem 3

We first provide some intuition about the strength of convergence of LLM-Polygon.

#### A.4.1 State Evolution and Markov Structure

At iteration $t$, let $G_t$ and $B_t$ denote the sets of good and bad views among the retained set $\mathcal{I}_t \subseteq \mathcal{I}_0$, and let

$$b_t = |B_t|$$

denote the number of bad views at iteration $t$. The algorithm always removes exactly one view, so

$$|\mathcal{I}_{t+1}| = |\mathcal{I}_t| - 1.$$

In LLM-Polygon, the algorithm proceeds by:

1. Assigning each view $I \in \mathcal{I}_t$ to one or more polygon faces $F \in \mathcal{F}$ and forming per-face embeddings $\mathbf{f}_{F,t}$ via CLIP feature averaging (or uncertainty vectors for unobserved faces).

2. For each $I \in \mathcal{I}_t$, computing a spatially grounded similarity score

$$s_I^{(t)} = \cos\big(f_{\mathrm{CLIP}}(I), \mathbf{f}_{F(I),t}\big),$$

   where $F(I)$ is the face (or chosen face) associated with $I$.

3. Deterministically selecting the *representative* and *ambiguous* views

$$I_{\mathrm{rep},t} = \arg\max_{I \in \mathcal{I}_t} s_I^{(t)}, \qquad I_{\mathrm{amb},t} = \arg\min_{I \in \mathcal{I}_t} s_I^{(t)}.$$

4. Querying the LLM with $\{I_{\mathrm{rep},t}, I_{\mathrm{amb},t}\}$ (and camera poses) to obtain a refined hypothesis $h_{t+1}$.

5. Removing from $\mathcal{I}_t$ whichever of $I_{\mathrm{rep},t}, I_{\mathrm{amb},t}$ has lower similarity to $h_{t+1}$ in CLIP space, yielding $\mathcal{I}_{t+1}$.

Thus, conditional on the current state (which includes $\mathcal{I}_t$, the per-face embeddings, and $h_t$), the choice of $I_{\mathrm{rep},t}, I_{\mathrm{amb},t}$ is *deterministic*; the only randomness comes from the LLM's internal sampling when producing $h_{t+1}$.

Consequently, the full state process $\{S_t\}_{t \geq 0}$ (where $S_t$ collects $\mathcal{I}_t$, the polygon structure, and $h_t$) is a Markov chain, and so is the derived process $\{b_t\}_{t \geq 0}$.

The key transition probabilities can be summarized as follows. There exist constants $\beta_{\mathrm{poly}}$ and $\alpha_{\mathrm{poly}}$ such that, whenever $b_t > 0$,

$$\Pr(b_{t+1} = b_t - 1 \mid b_t > 0) \geq \beta_{\mathrm{poly}}, \qquad \Pr(b_{t+1} = b_t + 1 \mid b_t > 0) \leq \alpha_{\mathrm{poly}},$$

with $\beta_{\mathrm{poly}} > \alpha_{\mathrm{poly}}$. Here

$$\beta_{\mathrm{poly}} = \Pr(\text{pair is good–bad}) \cdot \Pr(\text{LLM prefers good} \mid \text{good–bad pair}) \geq \delta_{\mathrm{poly}}\eta,$$

where $\delta_{\mathrm{poly}}$ is the probability (under the current state distribution) that the deterministically chosen pair $(I_{\mathrm{rep},t}, I_{\mathrm{amb},t})$ is a good–bad pair, and $\eta > 1/2$ is the semantic dominance constant.

Because polygonal spatial partitioning and per-face averaging bias $I_{\mathrm{rep},t}$ toward good, representative views and $I_{\mathrm{amb},t}$ toward ambiguous or inconsistent views, we obtain

$$\delta_{\mathrm{poly}} \gg \delta_{\mathrm{sample}},$$

where $\delta_{\mathrm{sample}}$ is the analogous probability under a baseline that selects pairs in an unstructured way. Thus, LLM-Polygon has a strictly larger drift toward eliminating bad views than LLM-Sampling.

**Proof 3** *We prove that under the stated assumptions, the number of bad views $b_t$ decays exponentially fast in $t$.*

**State and evolution.** *At iteration $t$, the algorithm maintains a retained set $\mathcal{I}_t \subseteq \mathcal{I}_0$ and an LLM hypothesis $h_t$. We define*

$$G_t = G \cap \mathcal{I}_t, \qquad B_t = B \cap \mathcal{I}_t, \qquad b_t = |B_t|.$$

*The LLM-Polygon update at iteration $t$ proceeds as follows:*

1. *Using polygon-based assignment and per-face CLIP feature averaging, compute per-face embeddings $\mathbf{f}_{F,t}$ and scores $s_I^{(t)}$ for each $I \in \mathcal{I}_t$:*

$$s_I^{(t)} = \cos\big(f_{\mathrm{CLIP}}(I), \mathbf{f}_{F(I),t}\big).$$

2. *Deterministically select*

$$I_{\mathrm{rep},t} = \arg\max_{I \in \mathcal{I}_t} s_I^{(t)}, \qquad I_{\mathrm{amb},t} = \arg\min_{I \in \mathcal{I}_t} s_I^{(t)}.$$

3. *Query the LLM with $\{I_{\mathrm{rep},t}, I_{\mathrm{amb},t}\}$ (and auxiliary context) to obtain a refined hypothesis $h_{t+1}$.*

4. *Remove from $\mathcal{I}_t$ whichever of $I_{\mathrm{rep},t}, I_{\mathrm{amb},t}$ has lower similarity to $h_{t+1}$ in CLIP space, yielding $\mathcal{I}_{t+1}$.*

*Thus at each iteration exactly one view is removed, and so*

$$|\mathcal{I}_{t+1}| = |\mathcal{I}_t| - 1, \qquad b_{t+1} \in \{b_t, b_t - 1\}.$$

*Let $S_t$ denote the full algorithmic state at time $t$, including $\mathcal{I}_t$, the polygon structure, and $h_t$. All steps in the transition $S_t \mapsto S_{t+1}$ are deterministic functions of $S_t$, except for the LLM output $h_{t+1}$, whose randomness depends only on its current query (itself determined by $S_t$). Therefore the process $\{S_t\}_{t \geq 0}$ is a Markov chain, and since $b_t$ is a measurable function of $S_t$, the process $\{b_t\}_{t \geq 0}$ is also Markov.*

**Lower bound on the probability of removing a bad view.** *Fix an iteration $t$ with $b_t > 0$. By Assumption (2) (geometric exposure of bad views), we have*

$$\Pr(\{I_{\mathrm{rep},t}, I_{\mathrm{amb},t}\} \cap B_t \neq \varnothing \mid S_t) \geq \delta_{\mathrm{poly}}. \tag{12}$$

*That is, with probability at least $\delta_{\mathrm{poly}}$, the deterministically selected pair contains at least one bad view.*

*On the event*

$$E_t := \big\{ \{I_{\mathrm{rep},t}, I_{\mathrm{amb},t}\} \cap B_t \neq \varnothing \big\},$$

*there are two possibilities:*

(a) *The pair is good–bad: $\{I_{\mathrm{rep},t}, I_{\mathrm{amb},t}\} = \{I_g, I_b\}$ with $I_g \in G_t$, $I_b \in B_t$.*

(b) *The pair is bad–bad: both $I_{\mathrm{rep},t}$ and $I_{\mathrm{amb},t}$ lie in $B_t$.*

*In either case, at least one bad view is present. When the pair is good–bad, Assumption (1) (semantic dominance) states that*

$$\Pr\big[\cos\big(h_{t+1}, f(I_g)\big) > \cos\big(h_{t+1}, f(I_b)\big) \,\big|\, \{I_{\mathrm{rep},t}, I_{\mathrm{amb},t}\} = \{I_g, I_b\}, S_t\big] \geq \eta.$$

*Since the algorithm removes the less similar view, the bad view $I_b$ is removed with probability at least $\eta$ in this case. In the bad–bad case, some bad view is removed with probability $1$, which only improves the chance of decreasing $b_t$. Hence, in all cases on $E_t$,*

$$\Pr\big(b_{t+1} = b_t - 1 \mid S_t, E_t\big) \geq \eta.$$

*Combining with equation 12, we get the uniform lower bound*

$$\Pr\big(b_{t+1} = b_t - 1 \mid S_t, b_t > 0\big) \geq \Pr(E_t \mid S_t, b_t > 0) \ \Pr\big(b_{t+1} = b_t - 1 \mid S_t, E_t\big)$$
$$\geq \delta_{\text{poly}} \cdot \eta.$$

*Define*

$$\beta_{\text{poly}} := \delta_{\text{poly}} \eta > 0.$$

*Then whenever $b_t > 0$,*

$$\Pr\big(b_{t+1} = b_t - 1 \mid S_t\big) \ \geq \ \beta_{\text{poly}}. \tag{13}$$

**Upper bound on the probability of removing a good view.** *We next upper bound the probability that a good view is removed at iteration $t$ while bad views still remain.*

*Whenever the pair is good–bad, the semantic dominance assumption implies that the probability of incorrectly removing the good view is at most $1 - \eta < \frac{1}{2}$. When the pair is good–good or bad–bad, removing a good view can only happen in the good–good case. Aggregating these possibilities, there exists a constant $\alpha_{\text{poly}} < \beta_{\text{poly}}$ such that for all $t$ with $b_t > 0$,*

$$\Pr\big(\text{remove a good view at step } t \mid S_t\big) \ \leq \ \alpha_{\text{poly}}. \tag{14}$$

*(If the algorithm never removes a good view as long as bad views remain, one may take $\alpha_{\text{poly}} = 0$; in general we only need that $\alpha_{\text{poly}} < \beta_{\text{poly}}$.)*

**Reduction to a two-state chain.** *Define the binary indicator*

$$X_t = \mathbf{1}\{b_t = 0\} = \begin{cases} 1, & b_t = 0 \quad (\text{no bad views remain}), \\ 0, & b_t > 0 \quad (\text{at least one bad view remains}). \end{cases}$$

*Then $\{X_t\}$ is a Markov chain on $\{0, 1\}$.*

*When $X_t = 0$ (i.e., $b_t > 0$), equation 13 implies*

$$\Pr(X_{t+1} = 1 \mid X_t = 0) = \Pr(b_{t+1} = 0 \mid b_t > 0) \ \geq \ \beta_{\text{poly}},$$

*so*

$$\Pr(X_{t+1} = 0 \mid X_t = 0) \leq 1 - \beta_{\text{poly}}.$$

*When $X_t = 1$ (no bad views remain), by interpreting any later reintroduction of bad views as a "corruption" event and using equation 14, we have*

$$\Pr(X_{t+1} = 0 \mid X_t = 1) \leq \alpha_{\text{poly}}, \qquad \Pr(X_{t+1} = 1 \mid X_t = 1) \geq 1 - \alpha_{\text{poly}}.$$

*Let $p_t = \Pr[X_t = 0] = \Pr[b_t > 0]$. Then*

$$\begin{aligned} p_{t+1} &= \Pr[X_{t+1} = 0] \\ &= \Pr(X_{t+1} = 0 \mid X_t = 0) \Pr(X_t = 0) + \Pr(X_{t+1} = 0 \mid X_t = 1) \Pr(X_t = 1) \\ &\leq (1 - \beta_{\text{poly}}) p_t + \alpha_{\text{poly}} (1 - p_t) \\ &= \alpha_{\text{poly}} + \big(1 - \alpha_{\text{poly}} - \beta_{\text{poly}}\big) p_t. \end{aligned}$$

*Define*

$$q := 1 - \alpha_{\text{poly}} - \beta_{\text{poly}}.$$

*Since $0 \leq \alpha_{\text{poly}} < \beta_{\text{poly}} \leq 1$, we have $-1 < q < 1$ and hence $|q| < 1$. The recurrence becomes*

$$p_{t+1} \leq \alpha_{\text{poly}} + q p_t.$$

**Exponential decay.**  *The linear recurrence*

$$p_{t+1} \leq \alpha_{\text{poly}} + qp_t, \qquad |q| < 1,$$

*has the general solution*

$$p_t \leq p^* + (p_0 - p^*)q^t, \qquad p^* := \frac{\alpha_{\text{poly}}}{\alpha_{\text{poly}} + \beta_{\text{poly}}}.$$

*Because $|q| < 1$, there exists $c_2 > 0$ such that*

$$|q|^t = e^{t \ln |q|} = e^{-c_2 t}, \qquad c_2 := -\ln |q| > 0.$$

*Thus*

$$p_t \leq p^* + |p_0 - p^*|e^{-c_2 t} \ \leq \ c_1 e^{-c_2 t}$$

*for some constant $c_1 > 0$ (absorbing $p^*$ and $|p_0 - p^*|$ into $c_1$ if desired). Since $p_t = \Pr[b_t > 0]$, this proves that*

$$\Pr[b_T > 0] \ \leq \ c_1 e^{-c_2 T}.$$

**Logarithmic iteration complexity.**  *Finally, for any $\varepsilon \in (0, 1)$, taking*

$$T(\varepsilon) = \left\lceil \frac{1}{c_2} \log \frac{c_1}{\varepsilon} \right\rceil = O(\log(1/\varepsilon))$$

*guarantees $\Pr[b_T > 0] \leq \varepsilon$ for all $T \geq T(\varepsilon)$. Equivalently, with probability at least $1 - \varepsilon$ the algorithm has eliminated all bad views by time $T(\varepsilon)$. This completes the proof.*

## B  Hyperparameter Calibration

We calibrate the two key hyperparameters, the confidence threshold and uncertainty weight, using a held-out subset of 5k COCO images. Matching (image–correct label) and non-matching (image–incorrect label) pairs are used to compute CLIP similarities, with distributions estimated via kernel density estimation (KDE). As shown in Figure 5, the distributions are well-separated (Cohen's $d = 5.06$).

The confidence threshold is set to the mean similarity of matching pairs ($\mu_+ = 0.311$, rounded to 0.32), and the uncertainty weight is set to the intersection of the distributions (rounded to 0.2), corresponding to the Bayes-optimal decision boundary.

### B.1  Hyperparameters

We present the hyperparameters used in our experiments at each step of the workflow. All images are 1920x1080 to preserve high-level detail.

***DetectAndSegment***  YOLOE (Wang et al., 2025) in prompt-free mode is used to generate object bounding boxes. Detections with fewer than 5,000 pixels or confidence below 0.15 are discarded. Features are extracted using OpenCLIP (Cherti et al., 2023) ViT-H-14 with the "laion2b_s32b_b79k" weights, and segmentation is performed with Segment Anything 2 (Ravi et al., 2024).

***MergeObservations***  In single-object experiments, all detections are assumed to correspond to the same object and are merged. In multi-object experiments, observations are merged based on semantic (visual and textual) and spatial similarity. Visual semantic similarity is computed as the average CLIP features of detections (excluding removed ones from the RefineAndPropose step) with a threshold of 0.6. Textual similarity is the cosine similarity of CLIP-encoded current object labels, thresholded at 0.25. Spatial similarity is measured via point cloud overlap, with a threshold of 0.1 to allow minimal overlap. Objects exceeding all three thresholds are merged.

Figure 5: **CLIP Feature Space: Similarity Distributions.** The plot shows the kernel density estimates (KDE) of cosine similarity distributions for matching (image–correct label) and non-matching (image–incorrect label) pairs within the CLIP feature space, using a held-out subset of COCO. The green distribution represents matching pairs, and its mean ($\mu_+ = 0.311$) is used as the **confidence threshold** (dashed green line), rounded to 0.32. This ensures that accepted labels correspond to in-distribution confidence levels. The red distribution represents non-matching pairs. The intersection point of the two distributions, which corresponds to the Bayes-optimal decision boundary, is used to set the **uncertainty weight**, rounded to 0.2.

*RefineAndPropose*   For both experiments, we set the maximum number of inner-loop iterations between data collection steps to 3, the confidence threshold to 0.32, the uncertainty weight to 0.2, and the number of polygon faces in the spatial partitioning algorithm to 8.

## B.2   Evaluation

### B.2.1   Setting the Success Threshold

As LADR is fully open-vocabulary, direct comparison with ground-truth labels is insufficient: the LLM may propose synonyms, which should be accepted. Since CLIP is sensitive to lexical variations, we use a Sentence Transformer (Reimers & Gurevych, 2019) to evaluate label equivalence. The final similarity for each prediction is the maximum of its similarity to the class name or description. To convert similarities into success rates, we construct a small set of synonym and non-synonym pairs, compute their similarities in the Sentence Transformer feature space, and visualize the distributions using kernel density estimation (KDE). The results show clear separation: while matching pairs can occasionally fall below 0.5, non-matching pairs never exceed 0.5. Based on this, we adopt 0.5 as the default threshold for evaluating label correctness.

To provide a more nuanced view, we also evaluate success rates at multiple thresholds:

- **0.3:** Almost all word pairs are detected as synonyms, including weakly related or contextually distant ones.

- **0.5:** Serves as a baseline, capturing meaningful synonyms while avoiding unrelated pairs.

- **0.7:** Mostly multi-word phrases with strong semantic alignment; loosely related pairs are excluded.

- **0.9:** Captures nearly identical or identical pairs, useful for exact matches.

By reporting success rates at these thresholds, we provide a more detailed picture of the model's behavior across varying levels of semantic similarity, from broad synonym detection to nearly exact matches.

### B.2.2   Matching Detections to Ground Truth

To evaluate multi-object detections, we assign each ground-truth object to the best-matching prediction based on a semantic-spatial similarity score, computed as a weighted combination of label similarity and spatial

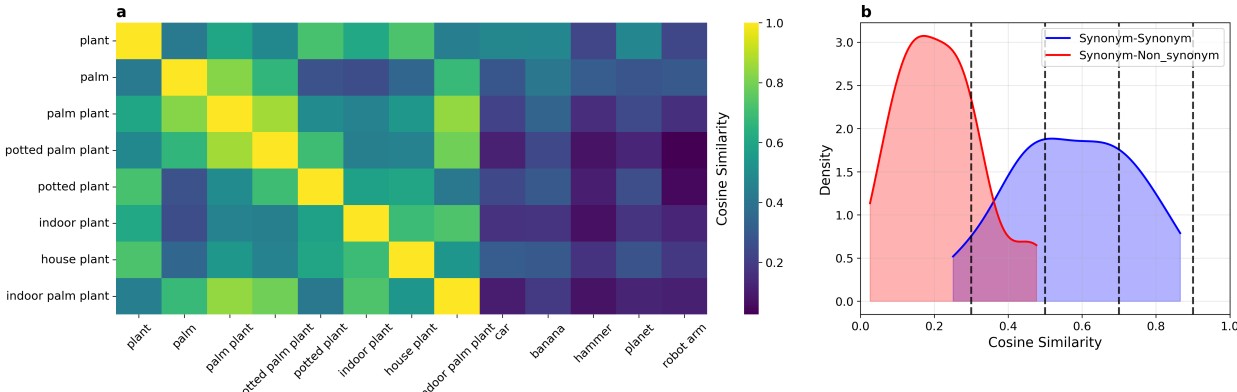

Figure 6: **Synonym Distance Analysis (a)** A cosine similarity heatmap for the word 'plant' and a set of related terms. The diagonal entries show high similarity for synonyms and near-synonyms (e.g., 'plant', 'potted plant', 'indoor plant'). Non-synonyms (e.g., 'car', 'banana', 'planet') exhibit low similarity. **(b)** Kernel density estimate (KDE) plots of cosine similarity distributions for synonym (blue) and non-synonym (red) pairs. The distributions show a clear separation, with a default threshold of 0.5 effectively distinguishing between the two. The dashed lines indicate various thresholds (0.3, 0.5, 0.7, and 0.9) used to evaluate the model's performance at different levels of semantic similarity, from broad synonym detection to near-exact matches.

overlap. Only matches with similarity above 0.1 are considered; lower values count as unsuccessful detections. Among eligible matches, the final assignment uses a bias-adjusted aggregation with phys_bias $= 0.2$ to select the best match.

### B.3 Datasets

### B.3.1 Single-Object Dataset

The single-object dataset comprises five instances for each of the five selected object classes from the OmniObjects3D dataset (Wu et al., 2023). These instances are used to generate multi-view image sequences, with representative examples shown in Figure 7.

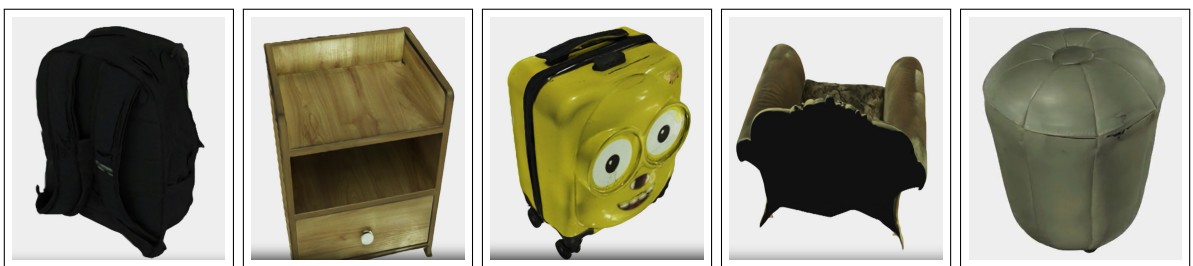

Figure 7: Examples of five object instances from the single-object experiments.

### B.3.2 Multi-Object Dataset

The multi-object dataset consists of custom 3D scenes created in NVIDIA Isaac Sim and manually labeled by the authors. To demonstrate the flexibility of our approach, we designed diverse environments using simulator-provided asset packs. The included room types are:

- **SimpleRoom:** open indoor spaces with a mix of miscellaneous objects,

- **Residential:** home-like settings with rug, chairs, and decorative items,

- **Commercial:** office area with a counter, a coffee-table and a storage unit,

- **Industrial:** warehouse-inspired space with shelving, crates, and utility equipment,

- **Vegetation:** outdoor theme featuring plants, trees, and garden elements.

Each scene contains multiple objects of interest, with dense arrangements to test robustness under occlusions, see Figure 8.

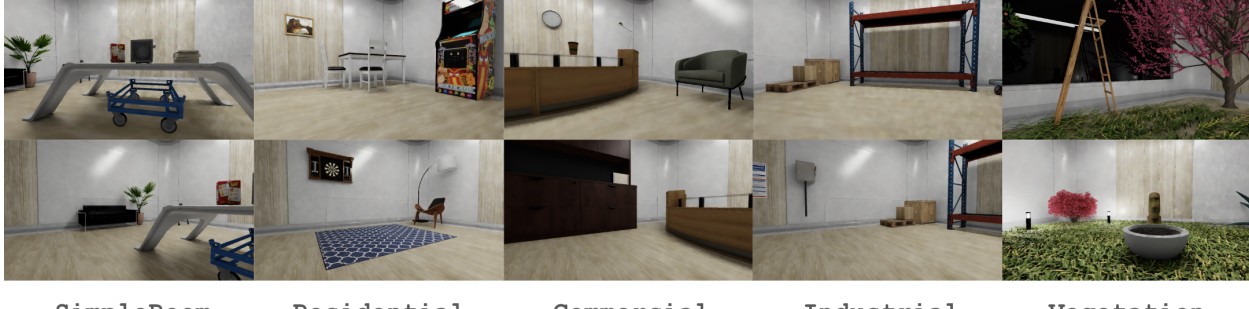

| SimpleRoom | Residential | Commercial | Industrial | Vegetation |

Figure 8: Five room scenes used in the multi-object experiments

### B.3.3 Experiment Configuration

In the single-object experiments, we average over 75 detections (5 classes × 5 instances × 3 seeds), starting with two initial views and allowing a budget of five additional views. In the multi-object setting, we average over 300 detections (5 scenes × 10 objects × 3 exploration policies × 2 seeds). At each position, eight new images are captured, beginning from a single initial position with a budget limit of three additional positions.

### B.4    Single-Object Experiment Results

### B.4.1    Detailed Results

| Algorithm | Class Sim | Desc Sim | Best Sim | Avg Sim | Succ@0.3 | Succ@0.5 | Succ@0.7 | Succ@0.9 | Avg Tokens |
|---|---|---|---|---|---|---|---|---|---|
| YOLO | 0.41 ± 0.25 | 0.31 ± 0.22 | 0.43 ± 0.26 | 0.36 ± 0.23 | 0.60 | 0.31 | 0.20 | 0.04 | 0 |
| CLIP | 0.51 ± 0.29 | 0.35 ± 0.20 | 0.52 ± 0.28 | 0.43 ± 0.23 | 0.64 | 0.39 | 0.31 | 0.16 | 0 |
| LLM-Label | 0.49 ± 0.28 | 0.38 ± 0.21 | 0.50 ± 0.28 | 0.42 ± 0.24 | 0.65 | 0.47 | 0.27 | 0.07 | 237 |
| LLM-Angle | 0.68 ± 0.31 | 0.59 ± 0.21 | 0.74 ± 0.27 | 0.64 ± 0.23 | 0.91 | 0.79 | 0.67 | 0.40 | 1575 |
| LLM-Tiled | 0.72 ± 0.27 | 0.62 ± 0.17 | 0.78 ± 0.23 | 0.67 ± 0.19 | 0.97 | 0.85 | 0.69 | 0.40 | 1008 |
| LLM-Random | 0.66 ± 0.26 | 0.62 ± 0.17 | 0.73 ± 0.21 | 0.64 ± 0.19 | 0.96 | 0.85 | 0.63 | 0.23 | 2182 |
| LLM-Sampling | 0.71 ± 0.30 | 0.63 ± 0.17 | 0.79 ± 0.23 | 0.67 ± 0.20 | 0.95 | 0.91 | 0.72 | 0.43 | 16115 |
| LLM-Polygon | 0.73 ± 0.24 | 0.66 ± 0.14 | 0.80 ± 0.17 | 0.69 ± 0.15 | 1.00 | 0.99 | 0.72 | 0.35 | 22176 |

Table 4: Detailed evaluation results for different algorithms. Similarity metrics are reported as mean ± standard deviation, followed by success rates at various thresholds and average LLM tokens used.

### B.4.2    Sample Results

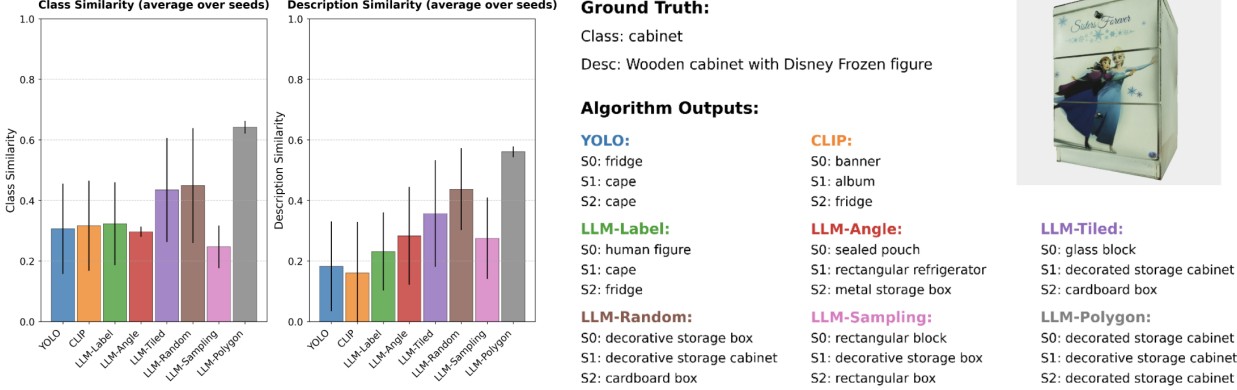

Figure 9: A per-object example showing algorithm performance. The bar charts on the left present class and description similarity, averaged over the seeds, while the right provides a qualitative example for an object in the 'cabinet' category from the single-object dataset. This example highlights that the generic 'cabinet' label is not sufficiently descriptive for this particular object.

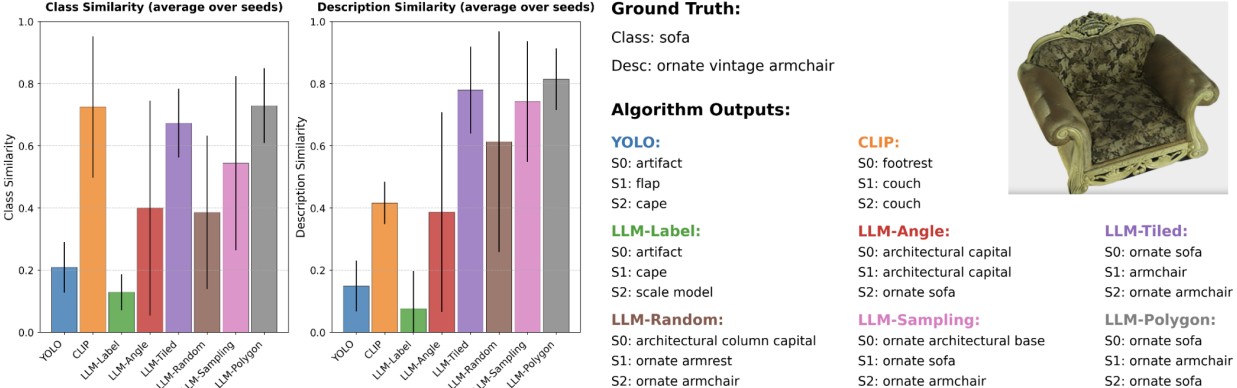

Figure 10: A per-object example showing algorithm performance. The bar charts on the left present class and description similarity, averaged over the seeds, while the right provides a qualitative example for an object in the "sofa" category in the single-object dataset.

## B.5 Multi-Object Experiment Results

### B.5.1 Detailed Results

| Algorithm | Class Sim | Desc Sim | Best Sim | Avg Sim | Succ@0.3 | Succ@0.5 | Succ@0.7 | Succ@0.9 | Avg Tokens |
|---|---|---|---|---|---|---|---|---|---|
| YOLO | 0.45 ± 0.24 | 0.34 ± 0.18 | 0.46 ± 0.24 | 0.39 ± 0.21 | 0.76 | 0.27 | 0.16 | 0.09 | 0 |
| CLIP | 0.51 ± 0.28 | 0.40 ± 0.23 | 0.52 ± 0.28 | 0.45 ± 0.24 | 0.77 | 0.40 | 0.28 | 0.16 | 0 |
| LLM-Label | 0.48 ± 0.26 | 0.38 ± 0.20 | 0.49 ± 0.25 | 0.43 ± 0.22 | 0.79 | 0.33 | 0.21 | 0.12 | 2965 |
| LLM-Angle | 0.56 ± 0.28 | 0.54 ± 0.28 | 0.63 ± 0.30 | 0.55 ± 0.26 | 0.82 | 0.62 | 0.47 | 0.26 | 8350 |
| LLM-Tiled | 0.57 ± 0.27 | 0.56 ± 0.28 | 0.64 ± 0.30 | 0.57 ± 0.26 | 0.82 | 0.62 | 0.47 | 0.25 | 6412 |
| LLM-Random | 0.56 ± 0.27 | 0.57 ± 0.28 | 0.64 ± 0.29 | 0.57 ± 0.26 | 0.80 | 0.63 | 0.48 | 0.26 | 12496 |
| LLM-Sampling | 0.59 ± 0.28 | 0.56 ± 0.28 | 0.65 ± 0.29 | 0.58 ± 0.26 | 0.84 | 0.64 | 0.50 | 0.29 | 14278 |
| LLM-Polygon | 0.59 ± 0.26 | 0.61 ± 0.28 | 0.67 ± 0.28 | 0.60 ± 0.25 | 0.86 | 0.69 | 0.55 | 0.27 | 17633 |

Table 5: Detailed evaluation results for different algorithms. Similarity metrics are reported as mean ± standard deviation, followed by success rates at various thresholds and average LLM tokens used.

### B.5.2 Sample Results

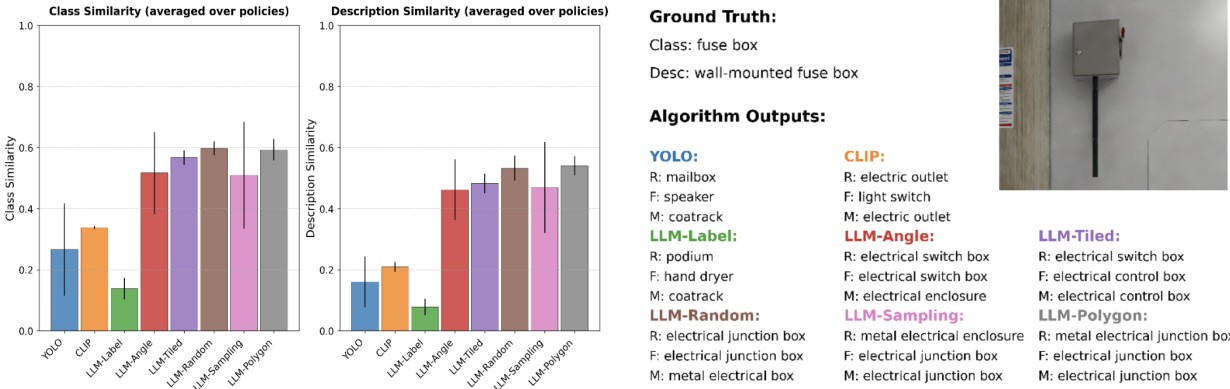

Figure 11: A per-object example showing algorithm performance. The bar charts on the left present class and description similarity, averaged over the exploration policies, while the right provides a qualitative example for an object with "fuse box" as the ground truth label in the multi-object dataset.

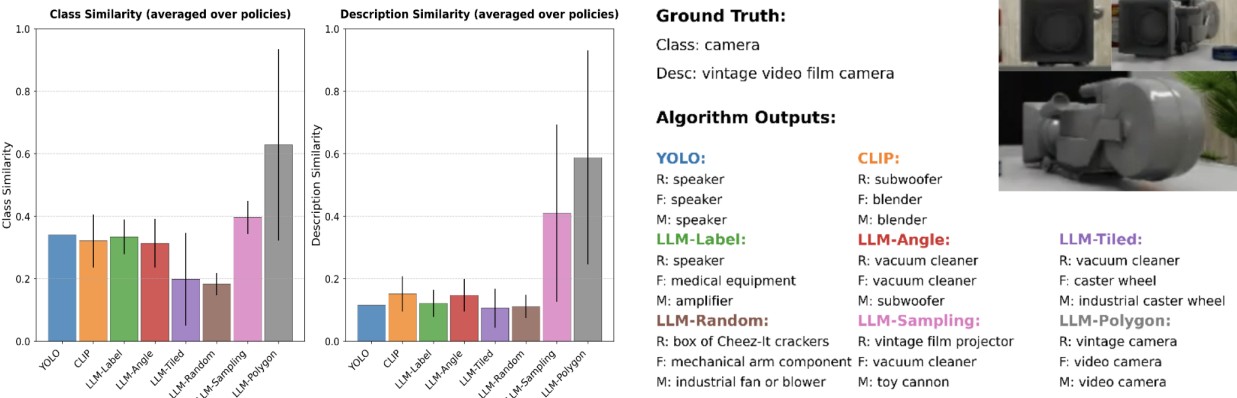

Figure 12: A per-object example showing algorithm performance. The bar charts on the left present class and description similarity, averaged over the exploration policies, while the right provides a qualitative example for an object with "camera" as the ground truth label in the multi-object dataset.

## B.6   LLM imagery input data

We provide examples of the LLM-Angle, and LLM-Tile in Figure 13.

## B.7   LLM Prompt

**Example of the prompt provided to the LLM for object labeling.** We provide the prompt used for LLM-Random. The prompts for other LADR algorithms are largely similar, with a few differences: neither LLM-Sampling nor LLM-Polygon requests a confidence score or the more descriptive view, and LLM-Polygon also does not request the next-best-view suggestion.

```
You will receive two images of the same object taken from (different) viewpoints, along with the angles (in degrees) from
↪    which they were captured. Analyze both images together considering their angles and return a single JSON object with
↪    these fields:

confident: true or false, indicating whether you are fully confident in the object's class based on the two views.
label: a brief class name of the object.
description: a clear, detailed description of the object for CLIP encoding. Focus on visually distinctive features (shape,
↪    material, color, texture, patterns) observable in at least one image.
next_best_angle: an integer in the range [-180, 180] suggesting the single most informative angle for revealing any ambiguous
↪    or missing features.
```

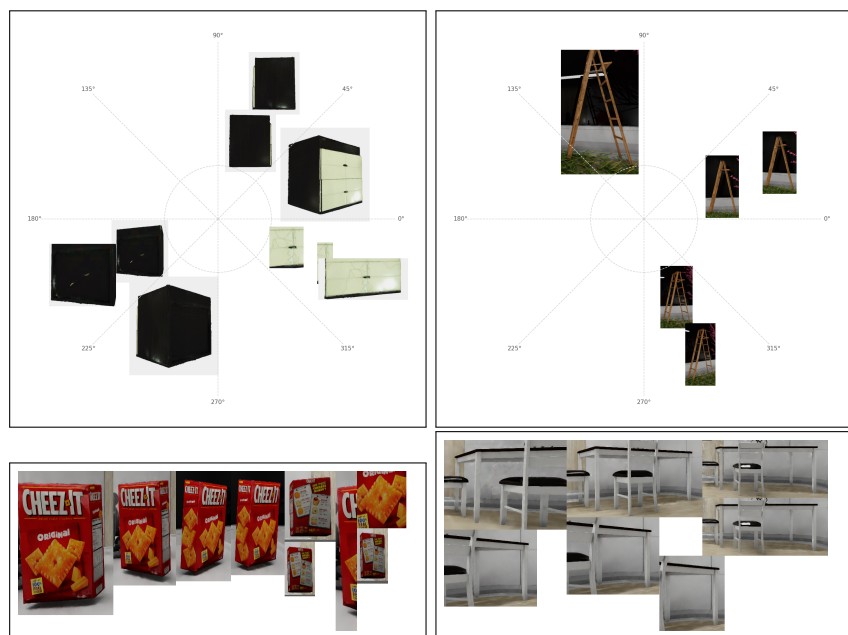

Figure 13: Top: input for LLM-Angle. Bottom: Input for LLM-Tile.

```
more_descriptive: either "left" or "right", indicating which image shows features most representative of the labeled class.
explanation: a short rationale covering:
    - why you set confident to true or false;
    - how you chose label and description;
    - why the proposed next_best_angle will improve clarity;
    - why the chosen image ("left" or "right") is more descriptive.

Guidelines:

Focus on the Main Object.
Each image is a crop around the objects bounding box, and the object fills most of the frame. Ignore background elements or
↪    smaller occluded items.

Combine Both Views and Angles.
Use both images and their provided angles to form a complete understanding. One view may reveal overall shape, while the
↪    other shows texture or details. Identify any remaining ambiguity or blind spots when choosing your next_best_angle.

Avoid Misidentifying from Partial Views.
If one image shows only a fragment (e.g., a handle), defer to the other image for overall class identification. Do not let a
↪    partial segment mislead your label.

Highlight Distinctive Features.
Describe only the most visually salient characteristics clearly visible in at least one image. Write in plain, factual
↪    language similar to alt-text or OpenCLIP-style captions.

Assess Confidence.
Set confident to true only if both images clearly support the same object class. If you suspect the label might change from
↪    another viewpoint or if one view is ambiguous, set confident to false and propose a next_best_angle that would resolve
↪    that ambiguity.

Determine "More Descriptive" View.
Compare the two images (left vs. right). Whichever one shows features most representative of the labeled classwhether by
↪    revealing overall shape, distinctive markings, or full extentshould be marked in more_descriptive. If both show equal
↪    detail, choose the one closest to the objects canonical appearance.

Next-Best-View Proposal.
Recommend a single integer angle in [-180, 180] that would most improve clarity of class or reveal missing features. Base
↪    your suggestion on the two given angles. For example, if the provided images are at 45 (left) and 60 (right), proposing
↪    0 might reveal the front; proposing 90 might reveal the opposite side.

Be Precise and Concise.
Write factually. Avoid speculation beyond what the two views suggest. Do not use generic class labels unsupported by the
↪    images.

Output Format
```

```
Return exactly one JSON object, for example:
{
  "confident": false,
  "label": "ceramic vase",
  "description": "a rounded ceramic vase with a narrow neck and blue floral patterns on a white background",
  "next_best_angle": 0,
  "more_descriptive": "right",
  "explanation": "The right image clearly shows the floral pattern and vase shape, but the left image only reveals the neck.
  ↪   Because the base is not visible from either 30 or 45, a 0 angle would show the full body and confirm the class."
}

Ensure that your JSON is valid, that all fields are present with the correct types, and that your response is accurate,
↪   well-structured, and concise.
Return only the raw JSON object.
```

