# OpenReview forum: "What One View Reveals, Another Conceals: 3D-Consistent Visual Reasoning with LLMs"
_TMLR — Accepted by TMLR_

### Review · Reviewer_iJ7t · 2026-01-28

**Summary Of Contributions:**

Contributions:
1. The paper proposes to incorporate LLM to generate view-consistent labels.
2. The paper introduces several variants, including LLM-Random, LLM-Sampling and LLM-Polygon, and validate the proposed algorithms across both single- and multiple-object settings.
3. The paper also provides theoretical analysis on the proposed variants.

Strengths:
1. The paper structure is clear and easy to follow.
2. Multiple variants are proposed and corresponding theoretical analysis is provided.

Weaknesses:
1. The paper claims to provide "rigorous Markov process-based analysis" (introduction), but the proof details are not defined. The markov process model is not clear (e.g., what is the state space? what is the transition model?). $X_s$ in (A1), $I_{\text{kill}, t}$, $\text{Bin}$ in (A2) are not well-defined.
2. The paper is somewhat motivated by "occlusions" in multiple-view detections (abstract, introduction, Sec.3 the 3D consistency problem), but did not shown any analysis/examples of how proposed models deal with occlusions. In Sec.4 the paper states "without loss of generality", but the reviewer considers multiple-object scenarios are more complicated as occlusions may exist.
3. The reviewer think "YOLO" and "CLIP" not valid baselines, as these have pre-defined label list. LLM-Random, or simply aggregate the LLM predicted labels on each image should serve as the naive baseline.
4. The authors are concluding the poor performance of "YOLO" and "LLM-Label" to lack of multi-view image baed reasoning (Sec. 6.4). The authors should consider evaluate the upper bound of these methods to further demonstrate the inferior performance does not come from the limited label list. For example, to use the most similar labels of the label list, what would be the performance.

**Audience:**

Yes

**Audience Explanation:**

The findings of this paper would benefit the researchers in the 3D community who work on object detection and recognition.

**Broader Impact Concerns:**

N.A.

**Claims And Evidence:**

No

**Claims Explanation:**

Please see weaknesses above. The reviewer considers the paper's claims on (i) rigorous analysis; (ii) occlusions and (iii) baselines & performance are not accurate and clear yet.

**Requested Changes:**

Please clarify and address the weaknesses mentioned above.
Also, there are some minor problems that require the clarification/revision from the authors:
1. It would be better to clarify the problem formulation clearly in section 3 or 4: What is the task input? What is the task output?
2.  Some figure/algorithm references are incorrect. For example at the end of page 4 there is a "Figure ??" and "Alg.1" does not exist in the paper.
3. In algorithm 4, it is not clear where do $s_{amb}$ and $s_{rep}$ come from (line 13). If they are just the CLIP score (line 6), isn't $s_{rep}$ always larger due to the $\text{argmax}$ in line 7&8?
4. Based on the reviewer's understanding, ConceptFusion mentioned in "YOLO" baseline did not aggregate YOLO labels as some final label.
5. In Figure 4, the largest threshold is 0.8, instead of 0.9.
6. As is mentioned in the conclusion, please provide a computational cost analysis (and with baselines), as the proposed method need to query LLM many times.

---

> ### Author Response · Authors · 2026-02-25
> **Response to reviewer comments**
>
> We thank the reviewer for his in depth comments and supportive view of our work. Below we repond to the reviewers comments.
>
> 1.	**The paper claims to provide "rigorous Markov process-based analysis" (introduction), but the proof details are not defined. The markov process model is not clear (e.g., what is the state space? what is the transition model?). $X_s$ in (A1), and $I_{kill,t}$  in (A2) are not well-defined.**
>
> Thank you for this important comment. In this revision we added section 5 on Markov process that should clarify the Markov framework, and specifically A1 and A2.
> We are happy to clarify any other questions.
>
> 2.	**The paper is somewhat motivated by "occlusions" in multiple-view detections (abstract, introduction, Sec.3 the 3D consistency problem), but did not shown any analysis/examples of how proposed models deal with occlusions.**
>
> First, we note that we also have occlusions happening in simulations environment.
> We now provide a real-data study which includes such occulsions (and other real-world challenges) and demostrate the ability of the baselines to address it.
>
> We note that the use of a VLM  allows to focus on each bounding box which helps in avoiding confusion between overlapping objects, occluded objects, etc.
>
> 3.	**The reviewer think "YOLO" and "CLIP" not valid baselines, as these have pre-defined label list. LLM-Random, or simply aggregate the LLM predicted labels on each image should serve as the naive baseline.**
>
> We accept this critic, and we changed the manuscript accordingly and noted those as closed-vocabs in section 7.1, for better transparency. Nevertheless we think that comparison is still informative to the audiance as it shows the sensitivity to confidence thrsholds as well as treatment of real-world scenarios by using open vs closed vocab.
>
> 4.	**The authors are concluding the poor performance of "YOLO" and "LLM-Label" to lack of multi-view image baed reasoning (Sec. 6.4). The authors should consider evaluate the upper bound of these methods to further demonstrate the inferior performance does not come from the limited label list. For example, to use the most similar labels of the label list, what would be the performance.**
>
> That essentially happens and we stated in 7.1 that we are using a Sentence transformer to consider synonim similarity for these baselines as well.
>
> **Requested Changes:**
>
> **It would be better to clarify the problem formulation clearly in section 3 or 4.**
>
> Done in section 3
>
> **Some figure/algorithm references are incorrect. For example at the end of page 4 there is a "Figure ??" and "Alg.1" does not exist in the paper.**
>
> Done
>
> **In algorithm 4, it is not clear where do $s_{amb}$ and $s_{rep}$ are coming from...**
>
> Thank you for catching this typo: a line was omitted in the previous version. Now in the revision the two vectors are the clip features that have the maximal and minimal cosine similarity with the current per-face average label.
>
> **Based on the reviewer's understanding, ConceptFusion mentioned in "YOLO" baseline did not aggregate YOLO labels as some final label.**
>
> If we undestand you correctly, we claim that our YOLO baseline is exactly what ConceptFusion do in their code, we average over the labels to get the final label.
>
> **In Figure 4, the largest threshold is 0.8, instead of 0.9.**
>
> Fixed
>
> **As is mentioned in the conclusion, please provide a computational cost analysis (and with baselines), as the proposed method need to query LLM many times.**
>
> We now provide in our revision a dedicated subsection 6.5 for the subject

---

### Review · Reviewer_4Gs7 · 2026-02-03

**Summary Of Contributions:**

This paper addresses the 3D consistency problem in open-vocabulary object detection. It aims to solve the issue of incorrect semantic labels introduced by varying viewpoints and occlusions. The proposed framework uses an iterative loop to sample informative views using CLIP features. It also uses LLM reasoning to generate and refine label hypotheses without a fixed label set. The paper provides a theoretical analysis showing the method achieves exponential coverage to a stable and correct semantic label. Experiments were conducted in simulated environment, and the results show that the proposed method achieves over 40% improvement in 3D semantic accuracy and sampling efficiency compared to standard YOLO and CLIP fusion.

Pros:
1. The paper provides a formal convergence analysis that proves the proposed method can lead to a correct label with high chance through its iterative pruning process (Theorem 3).
2. The paper inspected 3 variants of the proposed method and justify the value of the proposed components through ablation comparison.

Cons:
1. The proposed method requires multiple LLM queries for each object, which can significantly increase the inference latency and cost, limiting its real-world application.
2. The experiments were only conducted on a single dataset with NVIDA Isaac Sim simulation environment. Its efficacy on messy, real-world sensor data is questionable.

**Audience:**

Yes

**Audience Explanation:**

The proposed framework has some unique designs that might interest researchers in this area. The evaluation results on the simulation benchmark are also strong compared to the baseline approaches.

**Claims And Evidence:**

No

**Claims Explanation:**

On the bright side, the paper provides a solid mathematical grounding for the proposed method, and the polygon-based partitioning to guide LLM reasoning is a unique design. However, the experiments are lacking validation for real-world scenarios. Some parts of the idea, like introducing LLM reasoning into the process, share similarities to existing works not cited (see my comments in Requested Changes). Overall, I am on the fence but more on the decline side. I am open to change my mind if more evidence is provided on some more realistic benchmarks.

**Requested Changes:**

Some highly related works are not cited and compared in the paper. For example, OVODA [R1] also tackles the 3D open-vocabulary detection through a multi-view input, Scene-LLM [R2] proposes a 3D VLM that integrating the strong reasoning capability of LLM into 3D environment. The differences and similarity between these work should be discussed in the paper.

Ref:
- [R1] Xiang, Xinhao, et al. "Towards Open-Vocabulary Multimodal 3D Object Detection with Attributes." arXiv preprint arXiv:2508.16812 (2025).
- [R2] Fu, Rao, et al. "Scene-llm: Extending language model for 3d visual understanding and reasoning." arXiv preprint arXiv:2403.11401 (2024).

---

> ### Author Response · Authors · 2026-02-25
> **Response to reviewer comments and requests**
>
> We thank the reviewer for the helpful remarks, which have enabled us to improve the manuscript. We also appreciate the reviewer’s support of our theoretical analysis, the strength of our empirical results, and the ablation studies.
>
> We understand the reviewer’s concern regarding empirical validation on real-world data. In response, we conducted an additional real-world study using the ARKit dataset and included the results in Section 7.5. We hope this addition, together with addressing all other comments in the revision, resolves the reviewer’s concerns. Below, we provide detailed responses to each point raised:
>
> **Cons**
>
> **The proposed method requires multiple LLM queries for each object, which can significantly increase inference latency and cost, limiting its real-world applicability.**
>
> We now provide a more detailed runtime analysis per token in Section 6.5, including an evaluation of local small models that significantly reduce latency. That said, we acknowledge that smaller models may yield inferior performance unless carefully fine-tuned for the task.
>
> **The experiments were only conducted on a single dataset within the NVIDIA Isaac Sim simulation environment. Its efficacy on messy, real-world sensor data is questionable.**
>
> Motivated by the reviewer’s suggestion, we now include a detailed real-world study that highlights challenging conditions such as heavy occlusions, limited viewpoints, textureless surfaces (e.g., door vs. cabinet ambiguity), partial crops dominated by small objects (e.g., desk vs. mouse pad), missed detections, overlapping bounding boxes, and background distractors. We believe this extended evaluation provides a clearer understanding of the strengths and limitations of the proposed method and the different baselines.
>
> **Requested Changes**
>
> **Some highly related works are not cited and compared in the paper. For example, OVODA [R1] also addresses 3D open-vocabulary detection through multi-view input, and Scene-LLM [R2] proposes a 3D VLM that integrates the reasoning capabilities of LLMs into 3D environments. The similarities and differences between these works should be discussed.**
>
> **[R1] Xiang, Xinhao, et al. “Towards Open-Vocabulary Multimodal 3D Object Detection with Attributes.” arXiv:2508.16812 (2025).**
> **[R2] Fu, Rao, et al. “Scene-LLM: Extending Language Models for 3D Visual Understanding and Reasoning.” arXiv:2403.11401**(2024).
>
> We have added a discussion of these works in Section 2 and analyze them in relation to LADR, clarifying both conceptual similarities and methodological differences.

---

### Review · Reviewer_eFv4 · 2026-02-11

**Summary Of Contributions:**

This paper introduces LADR (LLM-guided Active Detection and Reasoning), a framework designed to address the problem of semantic label consistency in 3D object detection. The authors argue that existing zero-shot methods (combining detectors like YOLO with VLMs like CLIP) suffer from viewpoint bias, where non-descriptive or "bad" views dominate the aggregated feature representation. It provides a theoretical analysis modeling the refinement process as a Markov chain, proving exponential convergence to correct labels. Experiments in simulation (Isaac Sim, OmniObjects3D) demonstrate that LLM-Polygon significantly outperforms baselines (YOLO, CLIP, LLM-Tiled) in both single-object and multi-object scenarios.

**Additional Comments:**

I think it is a novel and complete work for TMLR. My recommendation is lean to accept.

**Audience:**

Yes

**Audience Explanation:**

1. Neuro-symbolic / Hybrid AI: The integration of Large Language Models (for reasoning) with traditional computer vision components (CLIP, YOLO) and geometric heuristics (spatial polygons)  is a highly relevant topic.

2. 3D Computer Vision & Robotics: The problem of "3D consistency" in open-vocabulary labeling is a persistent challenge in embodied AI. The proposed method for "active exploration"  is relevant to researchers working on autonomous agents and SLAM.

**Broader Impact Concerns:**

No major ethical concerns specific to this work.

**Claims And Evidence:**

Yes

**Claims Explanation:**

The paper presents LADR (LLM-guided Active Detection and Reasoning), a framework for resolving inconsistent semantic labels in 3D object detection by leveraging LLMs to refine hypotheses and actively sample views. The claims are supported by a combination of rigorous theoretical analysis and empirical evaluation in simulation.

Strengths in Evidence:

1. Theoretical Foundation: The authors provide a strong theoretical backing for their proposed algorithms (LLM-Random, LLM-Sampling, and LLM-Polygon). They model the label refinement process as a Markov chain and provide proofs for the exponential decay of "bad" (ambiguous or incorrect) views over time. This formalizes the intuition that active, spatially-grounded sampling yields faster convergence than random sampling.

2. Clear Motivation: The "piano" motivating example (Table 1) effectively demonstrates the failure modes of existing zero-shot fusion methods (like simple averaging of CLIP features) when "bad" views dominate the observation set. This convincingly establishes the problem statement.

3. Ablation and Progression: The experimental design isolates the contributions of each component. By stepping from LLM-Random (basic hypothesis testing) to LLM-Sampling (CLIP-guided) and finally to LLM-Polygon (spatial grounding), the authors provide clear evidence for the utility of geometric constraints in active perception.

**Requested Changes:**

1. Latency and Cost Analysis:

The paper mentions in the conclusion that the method requires "multiple inner-loop queries, which increases computational cost". However, there is no data quantifying this. Please include a table or discussion comparing the wall-clock time (inference latency) and token usage of LADR variants versus the baselines (YOLO/CLIP). This is crucial for readers to understand the trade-off between the reported "double digit improvement" in accuracy  and the computational overhead.

2. Sensitivity Analysis:

The method relies on specific hyperparameters, such as the confidence threshold (0.32) and uncertainty weight (0.2). While Appendix B discusses calibration, a sensitivity plot showing how performance degrades if these thresholds are suboptimal would be valuable. This would demonstrate whether the method is robust or requires precise tuning for every new environment.

3. Sim-to-Real Discussion:

Since experiments are simulation-only, please add a discussion (or a small qualitative experiment if feasible) regarding how the "Polygon" method handles real-world segmentation failures. Specifically, address what happens if the upstream DetectAndSegment  fails (e.g., under-segmentation merging two objects). Does the polygon wrapper force a single label on a multi-object cluster? A brief "Limitations" paragraph addressing this would suffice.

---

> ### Author Response · Authors · 2026-02-25
> **Respose to Reviewer Comments**
>
> We would like to thank the reviewer for his constructive comments and strong support of our works novelty, empirical and theoretical anaylsis and clarity. Below we answer the reviewers comments:
>
> 1. **Latency and Cost Analysis:** We provide in this revision analysis in section 6.5 where a comparison on different computation costs, from Yolo and two different VLMs with local and external deployment, different parameters size, and using an API from a remote server, (e.g. ChatGPT 5.2). We also quantify the running time per token.
>
> 2. **Sensitivity Analysis:** These thresholds are mostly for calibration and termination conditions, and show strong robustness across use cases. We infact show a sensitivty anaysis in Fig 3 as we change the threshold showing that sensitivty in accuracy to confidence threshold. As seen there the difference between the methods reamins however the success rate goes down with increasing the diffculty (higher threshold). Nevertheless, it is important to leave the level of certainty in the hands of the user, in our view, even as it trades off with the amount of sampling required to reach the prescribed confidence.
>
>     We saw in all experiments including the latest the arkit real-work scenes experiment that we added, that we can successfully reuse the same thresholds across different domains. We will highlight there how this is more of a problem between object types: eg objects with good texture will have higher confidence scores. This is difficult to mitigate without retraining the model.
>
> 3. **Sim-to-Real Discussion: Thank you for this important remark.**
> First, we note that we also have such issues happening in simulations environment we had so far. Our algorithms are leveraging existing detectors for initial object detections and bounding box fomration. Use of basic tools such as IoM criterion, is done on the initial detection. However, we see that in many cases during refinement the use of a VLM  allows to focus on each bounding box which helps in avoiding confusion between overlapping objects, occluded objects etc. In other words, the LADR algorithm is robust to a few mis-merged views: the LLM will throw these out, or flip to that object (assuming that these are the sides of the same object). So we are actually forcing a single object focus.
>
>     Second we now provide a real-world study with significant overlaps between objects (and other challenging modes) which demostrates the ability of the LLMs to overcome such problems.
>
>     We address this in our revision, as you requested, in section 4.

---

### Decision · Action_Editor_zggJ · 2026-03-24

**Recommendation:** Accept as is

**Audience:**

Yes

**Audience Explanation:**

The problem of achieving consistent semantic labeling across multiple views is an important challenge in 3D computer vision and embodied AI. The proposed framework integrates large language model reasoning with vision-based detection and active view selection, which reflects a growing research direction. The ideas presented in the paper may be useful for researchers working on open-vocabulary detection, 3D scene understanding, and robotic perception. The work provides useful insights and analysis that are likely to be of interest to the TMLR audience.

**Claims And Evidence:**

Yes

**Claims Explanation:**

This paper proposes LADR, a framework for improving semantic label consistency in multi-view 3D object detection using large language model reasoning and active view sampling. The work presents both a theoretical formulation and empirical evaluation to support the proposed approach. The manuscript has been revised following the review process to clarify aspects of the theoretical formulation and to provide additional experimental analysis and discussion. These revisions improve the clarity and completeness of the presentation. Overall, the technical approach is well motivated and the claims are supported by the evidence presented. The work makes a useful contribution to the area of multi-view visual reasoning. I recommend acceptance.